# Investigations into multiple fission yeast chromosome size determinants

Pei-Shang Wu[1],*, Todd Fallesen[2] and Frank Uhlmann[1],‡

## ABSTRACT

Mitotic chromosome dimensions differ between species, and they differ between developmental stages within an organism. The physiological determinants of chromosome size remain poorly understood. Here, we investigate chromosome size determinants in the fission yeast *Schizosaccharomyces pombe*. Super-resolution microscopy and semi-automated measurements reveal that cell and nuclear volume in interphase, and the time spent in mitosis (both previously proposed chromosome size determinants), have little influence on resultant chromosome dimensions. Instead, levels of the chromosomal condensin complex affect chromosome size, with increasing condensin levels resulting in more compact (thinner and shorter) chromosomes. Our observations inform the understanding of how chromosome dimensions are controlled in an organism. They suggest that a chromosome-intrinsic mechanism sets chromosome size, more so than the environment in which chromosomes find themselves in.

KEY WORDS: Chromosomes, Mitosis, Cell size, Condensin, *S. pombe*

## INTRODUCTION

Mitotic chromosome dimensions vary between eukaryotic species, which can harbour genomes of vastly different sizes (Flemming, 1882; Kramer et al., 2021; Sumner, 2003). Even in species with similar genome content, chromosome dimensions vary. For example, the budding yeast *Saccharomyces cerevisiae* and the fission yeast *Schizosaccharomyces pombe* harbour approximately similarly sized genomes, but *S. cerevisiae* has 16 short and thin chromosomes, whereas *S. pombe* has three long and thick chromosomes (Kakui et al., 2022). What defines the correct chromosome width in both species remains unknown. Another pair of organisms with similarly sized genomes are the closely related Chinese and Indian muntjacs. The DNA is distributed among 46 short and thin chromosomes in Chinese muntjacs, where Indian muntjacs have only six much longer and also much thicker chromosomes (Wurster and Benirschke, 1967,

1970). Again, how the Chinese and Indian muntjac chromosomes adopt their correct respective widths remains unknown.

Chromosome dimensions vary not only between species, but also within a species. For example, mitotic chromosomes appear sequentially shorter as cell size decreases during the cleavage divisions of early embryonic development in multicellular organisms (Hara et al., 2013; Zhou et al., 2023). The idea that chromosome size follows the size of the cell nucleus reaches back over 100 years ago, based on observations of sea snail early development (Conklin, 1912). In the nematode *Caenorhabditis elegans*, chromosomes also are relatively long at the one-cell stage, and they gradually shorten over the course of the first eight cleavage divisions. The increased DNA density in progressively smaller interphase nuclei has been suggested as the cause of correspondingly shorter and denser mitotic chromosomes (Hara et al., 2013). In another example, *Xenopus* chromosomes are also relatively long at the four-cell embryo stage, but they are shorter in the much smaller cells of a 4000-cell embryo towards the end of the cleavage division cycles. In this case, the decreasing cytoplasm-to-nuclear ratio, and consequently the reducing quantity of available cytoplasmic condensin I complexes, has been proposed as the reason for shorter chromosomes (Zhou et al., 2023).

Mitotic chromosome formation relies on a five-subunit protein complex, condensin, that belongs to the structural maintenance of chromosomes (SMC) family (Hirano, 2016; Uhlmann, 2016). Two types of condensin exist in vertebrates. Condensin I is a cytoplasmic protein complex, which gains access to chromosomes at the time of nuclear envelope breakdown in prophase. Condensin II in turn is nuclear, and also becomes enriched on chromatin between prophase and telophase (Gerlich et al., 2006; Hirota et al., 2004; Ono et al., 2004, 2003). The relative concentrations of condensin I and II affect the chromosome length-to-width ratio through an as-yet-unknown mechanism (Green et al., 2012; Shintomi and Hirano, 2011). Other organisms, including budding and fission yeasts, contain only one condensin complex (Saka et al., 1994; Sutani et al., 1999).

Condensin mediates mitotic chromosome formation by adding a layer of mitosis-specific long-range intra-chromosome interactions (Gibcus et al., 2018; Kakui et al., 2022, 2017; Tang et al., 2023). The range of these intra-chromosome interactions differs between species. A wider spacing between detectable condensin binding sites correlates with further-reaching interactions, as well as with wider chromosomes. Whether a causal relationship indeed links condensin spacing, chromatin interactions and chromosome dimensions, remains to be investigated. The absolute numbers of condensin molecules also differs between species, with around twice as many condensin complexes present per fission yeast genome, compared to budding yeast (Breker et al., 2013; Carpy et al., 2014). The effect of quantitatively different condensin levels on chromosome dimensions remains incompletely understood.

The dimensions of a mitotic chromosome arm of course depend on how much DNA is packed into it. Those chromosome arms that

[1]Chromosome Segregation Laboratory, The Francis Crick Institute, London NW1 1AT, UK. [2]Advanced Light Microscopy Science Technology Platform, The Francis Crick Institute, London NW1 1AT, UK.
*Present address: Laboratory for Cell Dynamics, Institute of Molecular Biology, Academia Sinica, Taipei, Taiwan.

‡Author for correspondence (frank.uhlmann@crick.ac.uk)

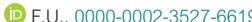 F.U., 0000-0002-3527-6619

contain larger genome portions are not only longer, they are also wider, both in yeasts and vertebrates (Kakui et al., 2022). Once formed, human chromosomes become progressively shorter and thicker the more time a cell spends in mitosis, and the width difference between short and long chromosome arms becomes increasingly prominent (Kakui et al., 2025). Human chromosomes are therefore not structures that are in a steady state. Rather, they appear to be approaching a steady state, but on biologically relevant timescales only the shortest chromosome arms might reach that state. To what extent time spent in mitosis affects chromosome dimensions in settings other than human cultured cells has not yet been documented.

Here, we systematically explore chromosome size determinants in the fission yeast *Schizosaccharomyces pombe*. We study the effects of cell and nuclear size, time spent in mitosis and of the amount of available condensin. To measure fission yeast mitotic chromosome dimensions, we develop an unbiased line scan algorithm. Our analyses rule out previously suggested parameters, namely, nuclear or cell size, and time in mitosis, as general chromosome size determinants. Instead, we find that condensin levels are a limiting determinant that defines chromosome dimensions in fission yeast.

## RESULTS
### Chromosome size in fission yeast cells of increasing size
We set out to explore whether the previously observed correlation between the size of the cell and nucleus in interphase and the chromosome size in mitosis, is a general relationship that also applies to fission yeast. To address this question, we utilized a fission yeast strain harbouring an ATP analogue sensitive *cdc2-asM17* allele (Aoi et al., 2014), allowing the use of the ATP analogue 1NM-PP1 to inhibit the CDK cell cycle kinase and impose cell cycle arrest in G2 phase. During the G2 arrest, fission yeast cells continue to grow, with nuclear size increasing at the same rate (Neumann and Nurse, 2007). We arrested cells in G2 for 3, 5 or 7 h by 1NM-PP1 treatment, then released them to progress into mitosis, where we blocked cell cycle progression again by transcriptional shut-off of the anaphase-promoting complex co-activator Slp1 (Kakui et al., 2017). This protocol resulted in cells with increasingly larger nuclei entering mitosis. As a comparison, we arrested cells in metaphase by Slp1 shut-off without any prior G2 arrest.

We measured cell size from bright-field images using image segmentation and cell masks generated by YeaZ (Dietler et al., 2020). The cell size in mitosis following 3 h G2 arrest was comparable to the size of untreated cells in mitosis, whereas cell size in mitosis after 7 h G2 arrest had more than doubled. To assess nuclear size, we tagged the nuclear periphery protein Uch2 with mCherry (Kouranti et al., 2010) and acquired fluorescent images in the TRITC channel, followed by image segmentation and nuclear mask generation using ilastik (Berg et al., 2019) (Fig. S1A). These analyses confirmed that, following G2 arrests of increasing durations, cells of increasing size, with increasingly larger nuclei, entered mitosis (Fig. 1A; Fig. S1B). Although we were unable to accurately measure nuclear size after 7 h G2 arrest due to an increasingly abnormal nuclear morphology. We also measured the chromatin occupied area in interphase nuclei of increasing sizes, using ilastik to create image masks of the DNA-binding dye DAPI-stained chromatin area. This analysis revealed that chromatin spread out over much greater areas in larger nuclei. The expanded interphase chromatin substantially contracted when cells with enlarged nuclei entered mitosis (Fig. 1A).

Following release from G2 arrest for 20 min into mitotic block due to Slp1 depletion, the majority of cells reached a mitotic state, indicated by the presence of short mitotic spindles (Fig. S1C). At this time, we imaged DAPI-stained mitotic chromosomes using

Airyscan superresolution microscopy (Huff, 2015; Kakui et al., 2022) (Fig. 1B). Z-stacks of images were acquired and projected. We then used a semi-automated tool to measure the width of chromosome arms that we traced on these images (Fig. S2A). The tool computationally straightens the traced chromosome arm, applies a sliding Gaussian fit along its length, and then records the average full width at half maximum intensity. This analysis revealed no significant changes to chromosome width, irrespective of the length of the G2 arrest, and consequently nuclear size, before entry into mitosis. The observation suggests that fission yeast mitotic chromosomes reach the same diameter irrespective of how widely distributed interphase chromatin is in nuclei of increasing sizes.

Next, we needed to measure chromosome arm lengths. Although we could trace chromosome portions and determine their widths on the images of metaphase-arrested cells, it was difficult to follow entire arms with enough certainty to establish their length. To measure chromosome arm lengths, we therefore changed our experimental approach. Instead of releasing G2-arrested cells into a mitotic block, we released cells to progress through mitosis and into anaphase, when we fixed and imaged cells as they segregated their chromosomes. At this time, chromosome arms were stretched out, and their length could be measured by tracing on Airyscan images (Fig. S2B). When we measured anaphase chromosome arm lengths following release from G2 arrests of increasing durations, we noticed a slight gradual lengthening, which reached statistical significance when comparing cells after 3 and 7 h of G2 arrest (Fig. 1C). These results suggest that chromosome arms are longer, but not wider, in larger fission yeast cells.

Our above result is consistent with a scaling mechanism in which fission yeast cell size influences the length of its chromosome arms in mitosis. At the same time, we must consider possible confounding factors for this conclusion, stemming from other differences between short and long cells. For example, chromosomes travel a further distance during anaphase in longer cells, and the accompanying movement might have straightened chromosome arms more in longer than in shorter cells. Straighter arms in turn would have made chromosomes appear longer in our two-dimensional measurements, when they might not in fact have been. Another confounding factor relates to the amount of available condensin in larger cells. When we analysed condensin levels in cells of increasing size by immunoblotting, we noticed that the relative condensin concentration decreased with increasing cell size, when compared to a metabolic housekeeping protein (G6PDH; Fig. 1D). Although the absolute number of condensin molecules is unlikely to have decreased in larger cells, a diluted condensin concentration in larger cells might nevertheless have reduced the efficiency of its chromosome association. We study the effect of an altered condensin concentration on chromosome size below. For now, to assess the merit of the concerns mentioned in this paragraph, we performed a reciprocal experiment, measuring chromosome dimensions in cells that are smaller than wild type.

### Chromosome size in fission yeast cells of a smaller size
Given the confounding factors around our observation that chromosome arms are longer in larger fission yeast cells, we performed a reciprocal experiment. We measured chromosome size in fission yeast cells that were smaller than wild type. If chromosome length scales with cell size, we would expect chromosomes to be shorter in smaller cells. For this, we employed two fission yeast cell size mutants in the cell cycle regulators Wee1 and PP2A. Cells lacking the CDK inhibitory kinase Wee1 (*wee1Δ*) or the major PP2A catalytic subunit (*ppa2Δ*) are smaller than wild-type cells (Fantes and Nurse, 1978; Kinoshita et al., 1993). Following the 5.5 h or 4.5 h that were

Journal of Cell Science

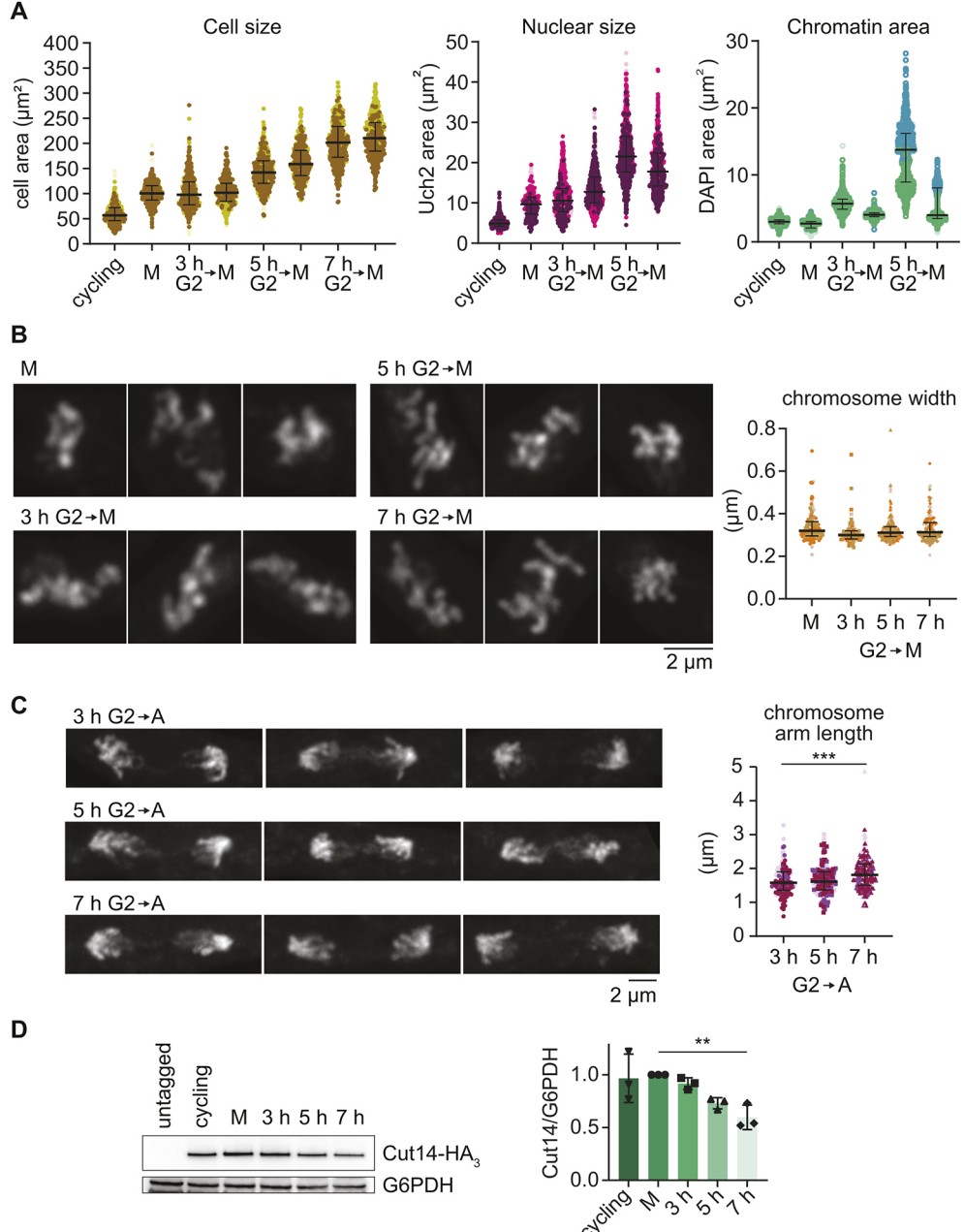

**Fig. 1. Chromosome size in larger cells.** (A) Cell size, nuclear size and chromatin areas were determined in cells arrested for increasing durations in G2, then released from G2 into mitotic arrest. Three biological repeat experiments were performed, colour coded, and results aggregated. The medians and interquartile ranges of the aggregated measurements are indicated (for the cell size experiment: cycling, *n*=615; M, *n*=686; 3 h G2, *n*=490; 3 h G2→M, *n*=535; 5 h G2, *n*=497; 5 h G2→M, *n*=459; 7 h G2, *n*=498; 7 h G2→M, *n*=507; for the nuclear size experiment: cycling, *n*=798; M, *n*=327; 3 h G2, *n*=796; 3 h G2→M, *n*=945; 5 h G2, *n*=940; 5 h G2→M, *n*=659; for the chromatin area experiment: cycling, *n*=1087; M, *n*=856; 3 h G2, *n*=1129; 3 h G2→M, *n*=1156; 5 h G2, *n*=1079; 5 h G2→M, *n*=902). (B) Representative Airyscan images of DAPI-stained chromosomes after G2 arrest for the indicated times and following release into mitotic block, as well as measured mitotic chromosome widths. Three biological repeat experiments were performed, colour coded, and results aggregated. The medians and interquartile ranges are indicated (M, *n* =115; 3 h G2→M, *n*=82; 5 h G2→M, *n*=163; 7 h G2→M, *n*=154). (C) Representative Airyscan images of DAPI-stained chromosomes following G2 release for the indicated times and release into synchronous anaphase progression, as well as measured chromosome arm lengths. Three biological repeat experiments were performed, colour coded, and results aggregated. The medians and interquartile ranges are indicated (3 h G2→A, *n*=142; 5 h G2→A, *n*=152; 7 h G2→A, *n*=225). ***P<0.0001 (one-way ANOVA with Tukey's multiple comparisons test). (D) Representative immunoblot analysing Cut14 levels at the indicated times in the G2 arrest, relative to G6PDH that served as the loading control. The Cut14/G6PDH ratios were quantified in three biological repeat experiments and normalised to the levels in the M population. Bars show the mean±s.d. **P=0.0070 (one-way ANOVA with Dunnett's multiple comparisons test).

required, respectively, to arrest both types of cells in mitosis by Slp1 depletion, both mutant cells and their nuclei remained substantially smaller compared to their wild-type counterparts (Fig. 2A; Fig. S3). We then measured chromosome width in the mitotically arrested

*wee1Δ* and *ppa2Δ* cells. Widths in *wee1Δ* cells did not differ from chromosome widths in similarly mitotically arrested control cells, whereas widths in *ppa2Δ* cells appeared slightly, but statistically significantly, reduced (Fig. 2B).

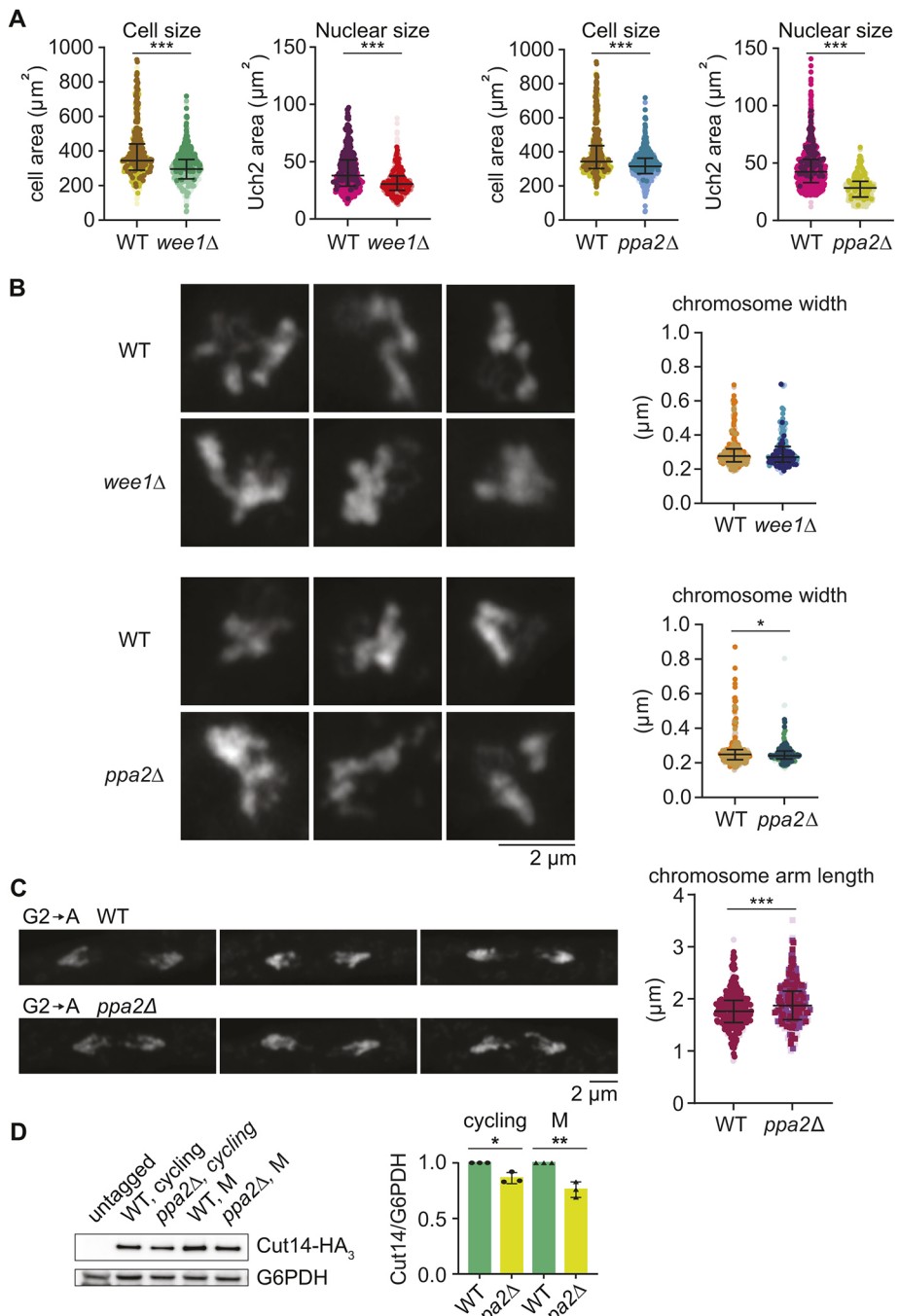

**Fig. 2. Chromosome size in smaller cells.**
(A) Cell size and nuclear size were determined in wild-type (WT) and *wee1Δ* cells after 5.5 h metaphase arrest due to Slp1 depletion. Three biological repeat experiments were performed, colour coded, and results aggregated. The medians and interquartile ranges are indicated (for the cell size experiment: WT, *n*=687; *wee1Δ*, *n*=771; for the nuclear size experiment: WT, *n*=418; *wee1Δ*, *n*=422). Similarly, WT and *ppa2Δ* cells were compared after 4.5 h of metaphase arrest (for the cell size experiment: WT, *n*=613; *ppa2Δ*, *n*=690; for the nuclear size experiment: WT, *n*=954; *ppa2Δ*, *n*=479). ***$P<0.0001$(unpaired two-tailed *t*-tests). (B) Representative Airyscan images of WT, *wee1Δ* and *ppa2Δ* DAPI-stained mitotic chromosomes at the respective arrest points, as well as measured chromosome widths. Three biological repeat experiments were performed, colour coded, and results aggregated. The medians and interquartile ranges are indicated (for chromosome width with *wee1Δ*: WT, *n*=276; *wee1Δ, n*=175; for chromosome width with *ppa2Δ*: WT, *n*=277; *ppa2Δ*, *n*=193). No significant difference for WT versus *wee1Δ*; *$P=0.0249$ for WT versus *ppa2Δ* (unpaired two-tailed *t*-tests). (C) Representative Airyscan images of DAPI-stained chromosomes following G2 arrest and release into synchronous anaphase of WT and *ppa2Δ* cells, as well as measured chromosome arm lengths. Three biological repeat experiments were performed, colour coded, and results aggregated. The medians and interquartile ranges are indicated (WT, *n*=422; *ppa2Δ*, *n*=361). ***$P<0.0001$ (unpaired two-tailed *t*-test). (D) Representative immunoblot to analyse Cut14 levels in WT and *ppa2Δ* cells, asynchronously growing or arrested in G2, relative to G6PDH that served as the loading control. The Cut14/G6PDH levels were quantified in three biological repeat experiments and normalised to the levels found in WT cells. Bars show the mean±s.d. *$P=0.0090$; **$P=0.0003$ (one-way ANOVA with Sidak's multiple comparisons tests).

We next turned to measuring chromosome arm length during anaphase in small fission yeast cells, following release from cell synchronisation in G2. We successfully obtained *ppa2Δ* cells containing the *cdc2-asM17* allele required for synchronisation by 1NM-PP1 treatment, but were unable to obtain *wee1Δ cdc2-asM17* cells. Against expectations, chromosome arms during anaphase of small *ppa2Δ cdc2-asM17* cells were not shorter, but longer, when compared to the wild-type control (Fig. 2C). This observation makes it unlikely that cell or nucleus size are directly linked to chromosome length. Rather, they suggest that a change occurred in both larger and smaller fission yeast cells that caused anaphase chromosomes to appear longer.

As undertaken above for larger cells, we measured condensin levels in *ppa2Δ* cells by immunoblotting. This analysis revealed a reduced condensin concentration in *ppa2Δ* cells when compared to wild-type cells, relative to the G6PDH housekeeping control protein (Fig. 2D). Although we do not know the reasons why condensin levels are reduced in these smaller cells, this observation opens the possibility that less condensin is available for mitotic chromosome formation in *ppa2Δ* cells compared to their wild-type controls, maybe resulting in their longer shapes. The second possible confounding factor that we considered in the case of larger cells, namely, a further separation distance during anaphase, is of course not applicable in smaller cells. Taken together, we find no consistent correlation between the size of the fission yeast cell and nucleus in interphase and the chromosome size in mitosis. Instead, we find that reduced relative condensin levels in both larger and smaller fission yeast cells correlate with somewhat longer chromosome arms.

## Condensin depletion causes wider and longer chromosomes

To test whether reduced condensin levels might be an underlying reason for longer chromosome arms in both larger and smaller fission yeast cells, we directly altered available condensin levels by tagging the condensin subunit Cut14 with an auxin-inducible degron (AID) tag (Kakui et al., 2017; Kanke et al., 2011). Titrating the auxin concentration that we added to the culture medium allowed gradual condensin depletion (Fig. 3A). At the same time as adding auxin, we supplemented the culture with thiamine to shut off Slp1 transcription and thereby induce mitotic arrest (Fig. S4A). After 5 h, mitotic chromosomes that formed in the presence of decreasing condensin concentrations were imaged by Airyscan microscopy. Even a small (∼30%) reduction of condensin levels caused by the addition of 25 µM auxin resulted in a measurable and significant widening of the resulting chromosomes, an effect that increased at 50 µM auxin, when Cut14 levels were less than half that of wild type (Fig. 3B). When condensin was further depleted (500 µM auxin) no measurable chromosomes could be seen in the mitotic arrest. These results demonstrate that condensin levels impact on chromosome width.

Next, we assessed the consequence of reduced condensin levels on chromosome length. We used our G2 synchronisation protocol and, at the same time as 1NM-PP1, we added auxin to the growth medium. To avoid complications from cell elongation, we again limited the G2 synchronisation period to 3 h, after which we released cells for progression into anaphase. Using this depletion protocol, 500 µM auxin addition resulted in an approximate halving of condensin levels. As a consequence, anaphase chromosome arms became significantly longer (Fig. 3C). These observations suggest that condensin levels impact on chromosome dimensions, with decreasing condensin levels resulting in both wider and longer chromosomes.

Above we observed that reduced condensin levels in both larger and smaller fission yeast cells correlated with longer, but not wider, chromosomes. A reason for this apparent discrepancy could be that chromosome length responds to reduced condensin levels more sensitively than width, or that the length changes are more readily measurable. Alternatively, factors additional to the condensin concentration might have affected chromosome dimensions in larger and smaller cells.

So far, we performed our chromosome measurements in fixed and stained whole-mount samples. An alternative common method to visualise chromosomes are chromosome spreads (Loidl and Lorenz, 2009). We therefore repeated the auxin-induced condensin depletion experiment in mitotically arrested cells and performed chromosome spreading, after enzymatic cell wall digestion and cell lysis. Chromosome width measurements yielded results that were quantitatively comparable to those obtained in whole-mount

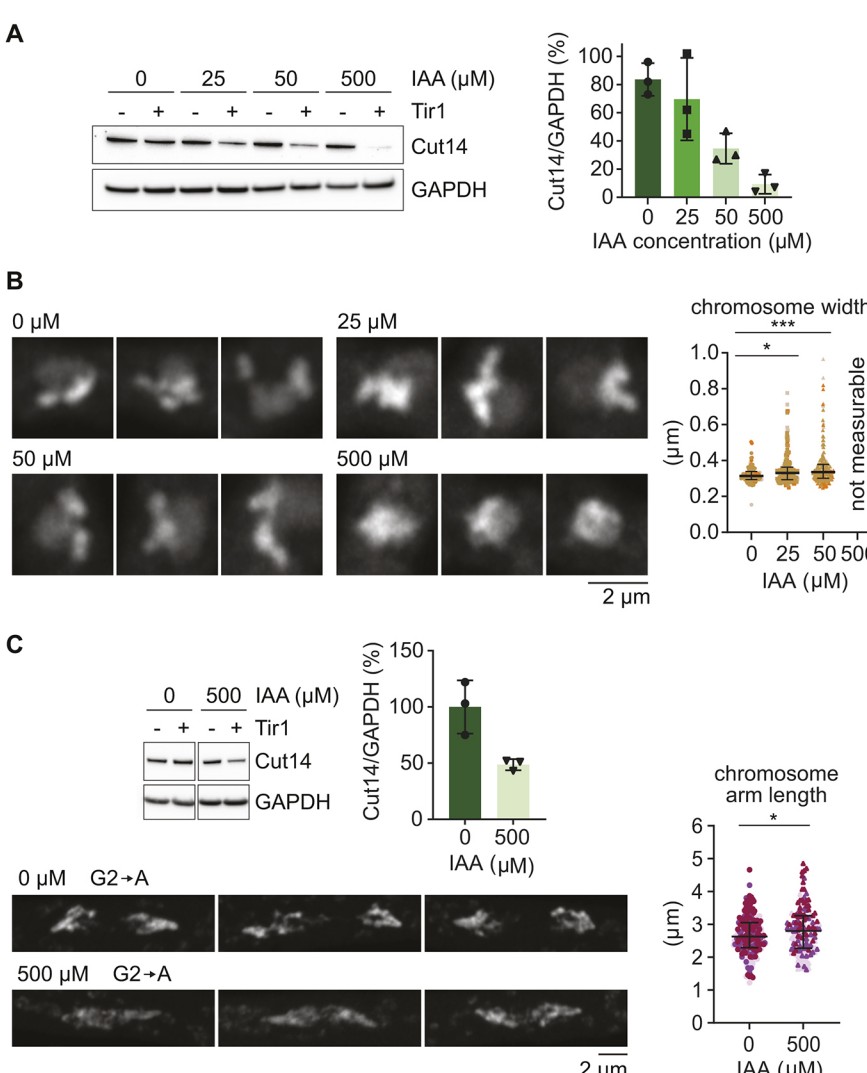

**Fig. 3. Chromosome dimensions following gradual condensin depletion.** (A) Representative immunoblot analysing Cut14 levels after 5 h depletion using the indicated auxin concentrations in Cut14-aid strains expressing, or not expressing, Tir1. GAPDH served as loading control. The Cut14/GAPDH ratios were quantified in three biological repeat experiments and normalised to the levels observed in the strain lacking Tir1. Bars show the mean±s.d. (B) Representative Airyscan images of DAPI-stained mitotic chromosomes in cells arrested and treated with the indicated auxin concentrations for 5 h, as well as measured mitotic chromosome widths. Three biological repeat experiments were performed, colour coded, and results aggregated. The medians and interquartile ranges are indicated (0 µM, n=100; 25 µM, n=158; 50 µM, n=183). *P=0.0296, ***P<0.001 (one-way ANOVA with Tukey's multiple comparisons tests). (C) Top left, a representative immunoblot to analyse Cut14 depletion is shown, as well as quantification relative to GAPDH that served as a loading control in triplicate biological repeat experiments. Bars show the means±s.d. Underneath and right, representative Airyscan images of DAPI-stained chromosomes following 3 h G2 arrest and then release into synchronous anaphase of Cut14-aid cells treated with 0 or 500 µM auxin, as well as measured chromosome arm lengths, three biological repeat experiments were performed, colour coded, and results aggregated. The medians and interquartile ranges are indicated (0 µM, n=209; 500 µM, n=144). *P=0.0177 (unpaired two-tailed t-test).

cells (Fig. S4B). We again observed widening chromosomes with increasing auxin concentrations, with no more discernible chromosomes visible at the highest concentration. However, we encountered slide-to-slide variation in measured chromosome widths, suggesting that the spreading procedure had the tendency to distort chromosomes. For subsequent experiments, we therefore returned to measuring chromosomes inside unperturbed whole-mount samples.

## Condensin and nuclear size in interphase

It has been suggested that condensin constrains interphase chromatin, thereby limiting its entropic expansion and thereby controlling nuclear size in both *Drosophila* and human cells (George et al., 2014). To examine whether such a role of condensin is conserved in fission yeast, we measured cell and nuclear size following condensin depletion (Fig. S4C). Against expectations, we did not observe larger nuclei in condensin-depleted cells. By contrast, we found that nuclei were somewhat smaller in mitotically arrested cells with reduced condensin levels. Cell size in mitosis was also smaller in condensin-depleted cells. We do not know the reason for reduced cell and nuclear size following condensin depletion, but cannot exclude that a role of condensin in transcriptional regulation might have contributed (Hocquet et al., 2018; Lancaster et al., 2021). We conclude that nuclear size, at least in fission yeast, is not limited by a role of condensin in constraining the interphase

chromatin volume. Rather, as we showed above, it is the nuclear size that limits how far interphase chromatin can spread (Fig. 1A).

## Extra condensin increases chromosome compaction

Chromosomes are wider and longer upon condensin depletion, consistent with the possibility that condensin levels regulate chromosome dimensions in fission yeast. However, larger chromosomes at reduced condensin levels could merely be an unphysiological intermediate on the way to chromosomes losing their shape owing to the lack of condensin (Fig. 3B) (Saka et al., 1994; Sutani et al., 1999). If condensin levels were indeed a limiting factor that determined fission yeast chromosome size, then increasing the available condensin concentration would be expected to result in smaller chromosomes. To investigate this possibility, we introduced an additional second copy of the genes encoding each of the five condensin subunits, Cut3, Cut14, Cnd1, Cnd2 and Cnd3, under control of their native upstream and downstream regulatory regions, at ectopic loci in the fission yeast genome (referred to as the '2nd copy' strain). The additional copies of each condensin gene resulted in a detectable condensin level increase, as seen by immunoblotting (Fig. 4A). Cell growth was mildly compromised by the additional condensin expression (Fig. S5A). Measuring chromosome dimension in mitotically arrested cells revealed distinctly thinner chromosomes in the 2nd copy strain as compared to the wild-type control (Fig. 4B). Similarly,

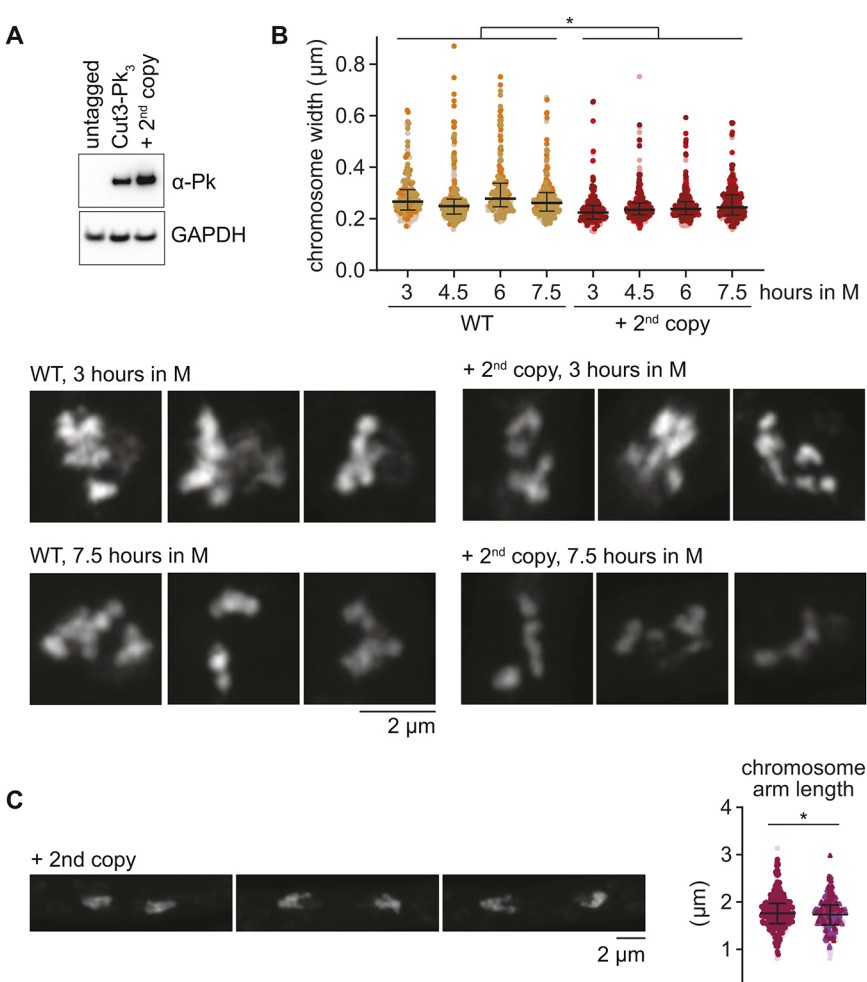

**Fig. 4. Chromosome dimensions at an increased condensin dosage.** (A) Immunoblot comparing endogenous Cut3-Pk$_3$ levels to those in the 2nd copy strain. An untagged Cut3 strain was included as control. GAPDH served as the loading control. Image representative of two repeats. (B) Representative Airyscan images of DAPI-stained mitotic chromosomes in WT and 2nd copy strains at the indicated times in a metaphase arrest, as well as measured chromosome widths. Three biological repeat experiments were performed, colour coded, and results aggregated. The medians and interquartile ranges are indicated (for the WT experiments: 3 h, $n$=176; 4.5 h, $n$=278; 6 h, $n$=207; 7.5 h, $n$=242; for the 2nd copy experiments: 3 h, $n$=208; 4.5 h, $n$=273; 6 h, $n$=284; 7.5 h, $n$=272). The time effect was not significant, *$P$=0.0353 for the genotype effect (two-way ANOVA). (C) Representative Airyscan images of DAPI-stained anaphase chromosomes in the 2nd copy strain, following 3 h G2 arrest and synchronous release, as well as measured chromosome arm lengths, compared to arm lengths in a WT strain. Three biological repeat experiments were performed, colour coded, and results aggregated. The medians and interquartile ranges are indicated (WT, $n$=422; 2nd copy strain $n$=343). *$P$=0.0304 (unpaired two-tailed $t$-test).

measuring chromosome arm length following release of G2 synchronised cells into anaphase revealed that chromosome arms were measurably shorter due to the increased condensin dosage (Fig. 4C). These observations demonstrate that condensin levels are indeed a limiting factor that defines chromosome dimensions during fission yeast mitosis. The greater the condensin levels the more compact (shorter and thinner) chromosomes become.

## Time in mitosis and fission yeast chromosome dimensions

Vertebrate chromosomes have been observed to undergo continuous shape changes over the course of an extended mitotic arrest, with chromosomes becoming gradually shorter and thicker over time (Kakui et al., 2025). Quantification of these shape changes showed that mitotic chromosomes are dynamic structures on the way to a steady state that only the shortest chromosome arms reach before anaphase onset. We therefore investigated whether fission yeast chromosomes similarly change shape in cells that are arrested in mitosis. To do so, we kept both a wild-type strain, as well as the 2nd copy strain, in a mitotic arrest due to Slp1 shut-off for increasing durations from 3 h up to 7.5 h (Fig. S5B). However, when we measured chromosome widths at successive times, we did not detect any measurable change (Fig. 4B). Furthermore, condensin levels remained constant during the duration of the arrest (Fig. S5C). These observations suggest that, unlike human chromosomes, fission yeast chromosomes reach a steady state faster, maybe due to their smaller size.

## Chromosome dimensions in unsynchronised cells

We have so far measured chromosome dimensions in synchronised or mitotically arrested cell populations. Finally, we asked whether these measurements are representative of chromosomes that form and are segregated during unperturbed cell divisions in asynchronously growing wild-type cells. Mitosis is only a short period of the cell growth and division cycle, and although this means anaphase cells are rare, the characteristic appearance of segregating chromosomes makes them easy to spot in an asynchronously growing cell population. When we measured the lengths of these anaphase chromosomes, we found them to be indistinguishable from chromosome lengths that we had measured in synchronised wild-type cells (Fig. 5A). Thus, the cell synchronisation treatments used in our study had not unduly altered anaphase chromosome lengths.

Identification of rare metaphase cells in an asynchronously growing fission yeast cell population, based on the DNA stain only, is more difficult. To identify cells in mitosis, we therefore utilised a tubulin–GFP fusion to visualise mitotic spindles. We selected nuclei that displayed short mitotic spindles as our targets for chromosome width measurements. The resultant measured chromosome widths were distinctly smaller (∼0.2 μm), compared to those that we previously measured in metaphase arrested cells (∼0.3 μm) (Fig. 5B). This difference could have arisen for two reasons. First, as metaphase is a very transient state during undisturbed mitosis, and the presence of a mitotic spindle cannot tell the exact time in mitosis, the chromosomes that we measured might have been pro-metaphase chromosomes that had not yet reached their final metaphase shape. Alternatively, it is possible that chromosomes indeed become wider during a mitotic arrest, as compared to during uninterrupted mitotic progression. The latter scenario bears resemblance to what we observed in human cells, where anaphase usually sets in before chromosomes reach a final steady state (Kakui et al., 2025). Live-cell super-resolution chromosome imaging might in the future be able to distinguish between these two possibilities.

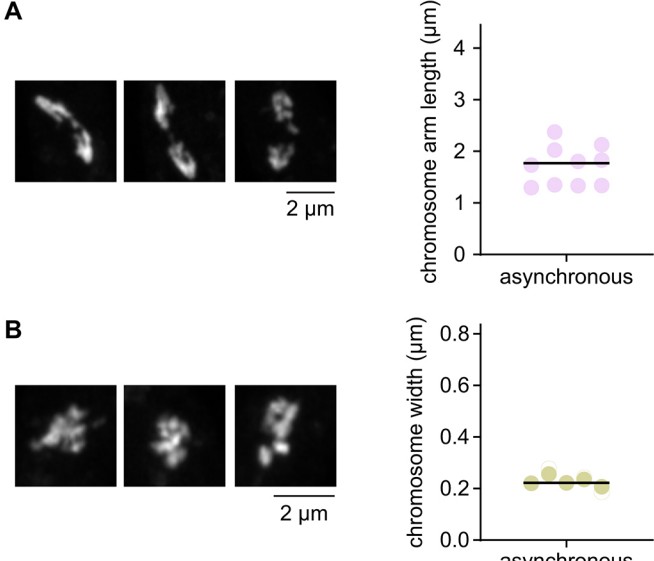

**Fig. 5. Chromosome size in unsynchronised cells.** (A) Representative Airyscan images of DAPI-stained chromosomes in anaphase in an asynchronously growing cell population. Anaphase chromosome arm lengths were measured and plotted. The line represents the median. (B) Representative Airyscan images of DAPI-stained chromosomes in asynchronously growing cells that displayed short mitotic spindles (visualised using a tubulin–GFP marker, not shown). Chromosome widths were measured and plotted. The line represents the median.

## DISCUSSION
### Cell size and chromosome size

Here, we performed a systematic study in the search for chromosome size determinants in fission yeast. To our surprise, we did not find a consistent correlation between cell or nuclear size in interphase and chromosome size in mitosis. Although chromatin was spread out over a far greater area in the larger nuclei of larger interphase cells, the ensuing mitotic chromosomes presented a width that was indistinguishable from that in wild type. However, in both larger and in smaller cells, we found that chromosome arms were measurably longer. This feature correlated with a reduced condensin concentration in both larger and smaller cells. Reduced condensin levels in large cells could stem from a dilution effect, where the absolute condensin amount remains constant but is diluted in a growing cellular environment. In contrast, it is not immediately obvious why condensin levels were reduced in small cells. Condensin has been reported to interact with PP2A in vertebrates (Takemoto et al., 2009). Whether a condensin–PP2A interaction exists in fission yeast that might have impacted on condensin levels in smaller *ppa2Δ* cells remains to be explored. How the levels of chromosomal proteins scale with cell size remains a topic of wider scientific interest (Schwaffer et al., 2021).

If longer chromosomes in both larger and smaller fission yeast are caused by reduced condensin levels, why did we not observe that chromosomes were also wider, as we did when we directly experimentally depleted condensin? It is possible that small changes to chromosome length are more readily discernible than small alterations in chromosome width. Alternatively, factors additional to reduced condensin levels could have contributed to the chromosome size changes that we observed in both larger and smaller fission yeast. We note that reduced condensin I levels were put forward as the reason for shorter, but not longer, chromosomes during *Xenopus* development (Zhou et al., 2023), the opposite of

the tendency that we observed in fission yeast. In the future, a quantitative assessment of chromosome-bound condensin, although experimentally challenging, should clarify whether less condensin is found on chromosomes in large and small fission yeast cells, and is the reason for the observed chromosome size changes.

## Chromosome shape changes over time

Unlike in vertebrates, where chromosomes keep changing shape for extended periods during a mitotic arrest (Gibcus et al., 2018; Kakui et al., 2025; Mora-Bermúdez et al., 2006; Shintomi et al., 2017), fission yeast chromosomes reach an apparently constant appearance in a relatively short time. Future work will need to establish whether fission yeast chromosomes have already reached their final state of compaction by the time cells enter anaphase under unperturbed conditions. At the latest, they reach their final appearance shortly after entering a mitotic arrest. Fission yeast condensin might work in ways different from vertebrate condensin, although we have no indication of overt molecular or structural differences. More likely, therefore, it is the case that the smaller fission yeast chromosome size allows them to reach a steady state faster. It will be interesting to establish the dimensions of this state, although the limitations of optical microscopy that we have encountered in our study make it hard to directly measure both the length and width of individual fission yeast chromosome arms. On visual inspection, the final fission yeast chromosome arm shape appears rod-shaped and elongated. This again contrasts the situation found in human chromosomes, where those short arms that reach a steady state display a rounded appearance (Kakui et al., 2025).

## Condensin levels and chromosome dimensions – loop capture or loop extrusion?

The biggest impact on fission yeast chromosome size that we could establish came from the cellular condensin levels. This conclusion finds support from previous studies using alternative methods to measure chromosome lengths, which also found that reduced condensin dosage results in longer chromosomes (Petrova et al., 2013; Saka et al., 1994; Schiklenk et al., 2018). Moreover, we demonstrate here that condensin is a rate-limiting component of chromosome compaction, with extra added condensin resulting in increased chromosome compaction beyond what is observed in wild-type cells. The more condensin, the shorter and the thinner chromosomes become. What can we learn from this behaviour about the mechanism of chromosome formation? Two prominent models for chromosome formation by condensin have been proposed in the recent literature, the loop capture and the loop extrusion models (Kim et al., 2023; Kinoshita et al., 2022; Tang et al., 2023; Uhlmann, 2025). In the loop capture model, condensin forms chromatin loops by sequential capture of DNA regions that come into contact by Brownian diffusion. This mechanism results in the formation of DNA rosettes that arrange themselves according to the principles of polymer physics in an elongated shape (Gerguri et al., 2021; Kakui et al., 2025). The loop extrusion model in turn foresees that condensin actively extrudes DNA loops until neighbouring condensins meet, thereby forming a chromosome backbone from which radial DNA loops emerge (Goloborodko et al., 2016; Samejima et al., 2025).

The loop extrusion model foresees that more condensins generate a greater number of loops along chromosomes, resulting in smaller loops and consequently thinner, but also longer, chromosomes. This prediction was not met by our experimental observations. Increased condensin levels did make chromosomes thinner, but also shorter. The loop capture scenario can accommodate the observations of

thinner and shorter chromosomes. If loop capture opportunities are unsaturated, additional condensin will create additional loops, which result in chromosome compaction in all dimensions, including both length and in width. In the future, characterising the loop interactions of condensin using genomics techniques like Hi-ChIP, dependent on the condensin concentration, will yield further insight into the mechanisms by which condensin shapes chromosomes.

## Limitations of this study

We used a simple eukaryotic model organism, the fission yeast *Schizosaccharomyces pombe*, for our search of chromosome size determinants. On the one hand, the genetic amenability of the organism allowed for facile control of cell size and cell cycle states. On the other hand, fission yeast chromosomes are small, and even super-resolution imaging approaches reached their limit. We were unable to simultaneously measure both the length and width of chromosomes, and we were unable to follow the changing chromosome shape in real time. An additional limitation of our study is that we could not distinguish the six individual fission yeast chromosome arms of different sizes. The measured size distributions were therefore wide, and observed differences in chromosome length or width were often small. We note, however, that a twofold chromosome volume difference relates to only a 1.26 (cubic root of 2)-fold size difference in each dimension. Despite these drawbacks, we were able to test the contribution of several cellular variables on resultant chromosome dimensions. In the future, investigations from a range of model organisms will combine their respective strengths and bring molecular insight into what the inside of a chromosome looks like and how chromosomes reach their iconic shape.

## MATERIALS AND METHODS

### *S. pombe* strains and culture

All strains used in this study are listed in Table S1. Strains were constructed using PCR-based gene targeting, and by genetic crossing and tetrad dissection. To construct the Cut14-aid strain, the $cut14^+$ gene was fused to the sequence encoding the auxin-inducible IAA17 degron module in a strain harbouring Skp1-TIR1 (Kakui et al., 2020). To obtain the 2nd copy condensin strain, three plasmids were created harbouring ectopic copies of the five condensin subunits (detailed in Table S2). These plasmids were linearised for integration at their respective marker loci in a host strain in which also the endogenous $cut3^+$ locus was fused to a $Pk_3$ epitope tag.

To achieve metaphase arrest, strains harbouring the $kan^R$::$P_{nmt41}$-$slp1^+$ allele (Kakui et al., 2017) were cultured in Edinburgh minimal medium (EMM; Petersen and Russell, 2016) supplemented with 2% glucose and 3.75 mg/ml glutamic acid, adenine, leucine, uracil and histidine supplements (Sigma) at 25°C overnight. The cells were filtered and resuspended in yeast extract (YE) medium (Petersen and Russell, 2016) containing 3% glucose and four amino acids supplements (YE4S; Petersen and Russell, 2016), plus 5 µg/ml thiamine (Sigma) to shut off Slp1 expression for the indicated times at 25°C. To induce Cut14 depletion in the $kan^R$::$P_{nmt41}$-$slp1^+$ background, the Cut14-aid strain was cultured as above, with auxin (Sigma) supplemented at the indicated concentrations at the time of shifting to thiamine-containing YE4S medium for 5 h at 25°C. To arrest cells in G2, strains harbouring the $cdc2$-$asM17$ allele (Aoi et al., 2014) were cultured in EMM4S medium at 30°C overnight. 1 µM 1-NM-PP1 (Adooq Bioscience) and 5 µg/ml thiamine were added the next day for 3, 5 or 7 h. To transition from G2 to a mitotic arrest, cells were filtered and washed at the indicated times and released into thiamine-containing YE4S medium for 20 min at 30°C. To release cells from G2 to anaphase, the $cdc2$-$asM17$ strains were treated as described above, except for omitting thiamine addition to the EMM4S or YE4S media. Samples were in this case collected at 2-min intervals.

### Chromosome width measurements

Collected cells were fixed with 70% ethanol. After fixation, the samples were resuspended in 10% glycerol containing 0.25 µg/ml DAPI. Mitotic

Journal of Cell Science

chromosomes were imaged by Airyscan microscopy (Huff, 2015) using a 63×/1.40 NA objective lens. Images were acquired along the z-axis with the first/last option and optimal spacing. Snapshots of mitotic spindles that span across the nucleus were taken to monitor the mitotic cell population. To measure chromosome width, chromosome arms were segmented using a custom Fiji macro (Schindelin et al., 2012). We selected chromosomes by placing points along the long chromosome axis on a maximum intensity projection of a 3D image stack using the 'polyline' tool. The line selection along the chromosome axis was straightened using the 'Straighten' tool in Fiji. The image was then cropped along the axis of the straightened chromosome and the region of interest (ROI) saved. The ROI was overlaid on the original image to ensure that each chromosome was selected only once.

Image analysis of the segmented chromosome arms was performed in MATLAB. Chromosome images were first aligned so that the chromosome long axis was aligned with the vertical axis of the image. For each chromosome, the image was thresholded and a binary mask created. Holes in the mask were filled, and a convex hull of the mask was generated. At this point, the mean width of the binary mask rows was measured, and any individual row that had a width less than 70% of the mean was removed. This step removes any telomere ends from the dataset. The first rows from each chromosome end where the width is greater than 70% of the mean are marked as border rows. The binary mask is then expanded by 2 pixels, and applied to the chromosome image, setting all background intensities to zero.

The image was then normalised, and a Gaussian fit applied to the image intensity in each row along the chromosome axis. The rows were then filtered, and any row with a $r^2$ less than the specified threshold or with a confidence interval that was an outlier as measured by the *rmoutliers* function, were removed from the dataset. The full width at half maximum (FWHM) value of each Gaussian was then found and compared to all the FWHMs for each row in the chromosome image. Any row with a FWHM deemed an outlier by the *rmoutliers* function was removed from the dataset. The remaining rows were then combined to give a mean measurement of the width for the chromosome, as well as standard deviation and standard error of the mean.

The mean FWHM and standard error of the mean are plotted for each chromosome dataset (i.e. multiple chromosomes under the same condition). A histogram of the mean FWHMs is also generated per dataset. A montage image for inspection is also created, where the rows that are measured are overlaid in blue on the original image, with the border rows in red. The chromosome width measurements were performed from triplicate biological repeat experiments and at least 50 nuclei were imaged for each experiment.

All custom image analysis tools in Fiji and MATLAB, with usage instructions, have been deposited with the Zenodo repository, where they can be accessed at doi:10.5281/zenodo.15657164.

### Chromosome arm length measurements
To measure chromosome arm lengths, we fixed and imaged cells as they segregated their chromosomes during anaphase, either following synchronous release from G2 arrest, or in asynchronously growing cell populations. DAPI-stained DNA was imaged using Airyscan microscopy, as described for chromosome width measurements. Chromosome arms were then traced from one end to the other with segmented lines. Only arms with clearly discernible ends were traced. The corresponding arm lengths were then recorded using the analysis option in Fiji. All measurable arms that point in the direction of anaphase separation were included in the measurements. As the DNA stain does not distinguish between the fission yeast chromosome arms, the measurements are an aggregate of the lengths of all six arms.

### Chromosome spreads
The chromosome spread protocol was adapted from previously published procedures (Dickinson and Isenberg, 1982; Flor-Parra et al., 2014; Loidl and Lorenz, 2009). Early exponential phase cultures were arrested in mitosis, as described above. Cells from 50 ml of culture were harvested by centrifugation (2000 *g* for 1 min) and resuspended in pretreatment buffer [100 mM 2-(N-morpholino)ethanesulfonic acid (MES)-NaOH pH 6.0, 100 mM EDTA and 5 mM DTT] with added 0.1% sodium azide. After two washes in pretreatment buffer, cells were resuspended in 5 ml pretreatment buffer and incubated at 30°C for 1 h with shaking. Cells were harvested by centrifugation (2000 *g* for 1 min) and washed once with 20 mM

MES-NaOH pH 6.0, 0.6 M sorbitol and twice with SCS buffer, prepared by mixing four volumes of SCSa (20 mM sodium citrate and 1 M D-sorbitol) with one volume of SCSb (20 mM citric acid and 1 M D-sorbitol), resulting in pH 5.8. After these washes, the pellet was resuspended in 6 ml SCS buffer containing 0.3 g/ml lallzyme MMX (IOC Myzym Elevage), 0.5 mg/ml zymolyase 100T (MP Biomedicals), 12 µl 1 M PMSF in DMSO and 30 µl 1 M DTT. The suspension was incubated at 37°C until almost all cells appeared rounded when inspected by phase-contrast microscopy. At this point, the suspension was split into two 15 ml centrifuge tubes and the spheroplasts were recovered by centrifugation at 700 *g* for 4 min and resuspended in 3 ml prewarmed stop solution A (100 mM MES-NaOH pH 6.4, 1 mM EDTA, 0.5 mM MgCl$_2$ and 1 M sorbitol). 75 mM KCl and 2 mM PMSF were added, and the suspension was incubated at 37°C for 15 min. Spheroplasts were then harvested again (900 *g* for 3 min), washed with stop solution A, and finally resuspended in 100 µl of stop solution A, to which 0.5 µl of 1 M PMSF was added. In a chemical fume hood, 20 µl cell suspension was dropped onto the middle of an acid-washed glass slide, followed by 40 µl fixative (4% paraformaldehyde supplemented with 3.6% sucrose) and 80 µl of 1% Lipsol. After 90 s, additional 80 µl of fixative was added. The mixture was then spread out with a glass rod, and the slide was then kept drying inside the fume hood overnight. The slide was mounted with 6 µl of 10% glycerol supplemented with 3 µg/ml DAPI. Slides were sealed with nail polish and imaged using a DeltaVision system built around an Olympus IX70 microscope with a 100× NA1.4 objective lens.

### Cell size, nuclear size and chromatin area measurements
Collected cells were fixed with 70% ethanol or 3.6% formaldehyde. Images of bright field, TRITC (visualising Uch2–mCherry), DAPI and FITC (visualising GFP–Atb2) channels were acquired with a DeltaVision or a Zeiss Observer microscope using a 63× objective lens. At least 200 cells were imaged in seven slices at 0.5 µm spacing along the z-axis for monitoring cell area, nuclear area, chromosome area and the population displaying mitotic spindles, respectively. The z-stacks were flattened by maximum intensity projection, and corresponding cell masks were created using YeaZ (Dietler et al., 2020); the nuclear and chromatin area masks were generated by ilastik (Berg et al., 2019). The masks were then inspected using Fiji. Objects that were not successfully masked, or did not display a mitotic spindle, were removed using the wand tool. Conversely, when measuring nuclear or chromatin areas in cycling cell populations, cells in metaphase that showed mitotic spindles were removed from the analysis. Subsequently, the look up table was converted into a colour range for thresholding, and the scale was converted from pixels to µm. The area was then measured by selecting the 'analyze particles' option. Triplicate biological repeat experiments were performed in all cases.

### Western blotting
Whole-cell protein extracts were prepared using the TCA extraction method (Kakui and Uhlmann, 2019), separated by 4–12% Tris-Glycine SDS-polyacrylamide gel electrophoresis and transferred onto nitrocellulose membranes. The following primary antibodies were used for detection: anti-HA (clone 12CA5, Roche, cat. no. 11666606001; 1:1000), anti-Pk (clone SV5-Pk1, Bio-Rad, cat. no. MCA1360; 1:1000), anti-GAPDH (clone GA1R, abcam, cat. no. ab125247; 1:10,000), anti-G6PDH (Merck, cat. no. A9521; 1:5000), followed by peroxidase coupled secondary antibodies. The blots were developed with enhanced chemiluminescent reagents and signals detected using a Bio-Rad ChemiDoc MP gel documentation system. Images acquisition was concluded before any signals reached saturation, and band intensities quantified using Image Lab software (Bio-Rad).

### Acknowledgements
We would like to thank Céline Bouchoux, Ying Gu, Thomas Hammond, Yasu Kakui, Sofi Mebrate, Snezhana Oliferenko, Matt Renshaw, Billy Whyte and Theresa Zeisner for reagents, help and advice, Rocco D'Antuono and Matt Renshaw from the Crick Advanced Light Microscopy STP and Gavin Kelly from the Bioinformatics & Biostatistics STP for help with microscopy and statistics, as well as all our laboratory members for discussions and comments on the manuscript.

### Competing interests
The authors declare no competing or financial interests.

## Author contributions
Investigation: P.-S.W., T.F., F.U.; Writing – original draft: P.-S.W., F.U.; Writing – review & editing: T.F.

## Funding
This work was supported by a Wellcome Trust Investigator Award (220244/Z/20/Z) and the Francis Crick Institute (cc2137), which receives its core funding from Cancer Research UK, the UKRI Medical Research Council, and the Wellcome Trust. Open Access funding provided by The Francis Crick Institute. Deposited in PMC for immediate release.

## Data and resource availability
All custom image analysis tools in Fiji and MATLAB, with usage instructions, have been deposited with the Zenodo repository, where they can be accessed at doi:10.5281/zenodo.15657164. All other relevant data and details of resources can be found within the article and its supplementary information.

## Peer review history
The peer review history is available online at https://journals.biologists.com/jcs/lookup/doi/10.1242/jcs.264569.reviewer-comments.pdf

## Special Issue
This article is part of the Special Issue 'Cell Biology of the Nucleus', guest edited by Abby Buchwalter. See related articles at https://journals.biologists.com/jcs/issue/139/12.

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
