## [Peer Review File · Journal of Cell Science]

Investigations into fission yeast chromosome size determinants

Pei-Shang Wu, Todd Fallesen and Frank Uhlmann

DOI: 10.1242/jcs.264569

Editor: Megan King

Review timeline

Submission to Review Commons:	13 September 2025
Submission to Journal of Cell Science:	5 November 2025
Editorial decision:	10 November 2025
First revision received:	5 January 2025
Accepted:	9 January 2025

Reviewer 1

Evidence, reproducibility and clarity

****Summary****

In this manuscript, Wu et al investigate the shape of mitotic chromosomes under normal conditions and after altering cell cycle parameters and condensin levels. By quantitatively measuring chromosome width and length, they show that chromosome shape in *S. pombe* is remarkably robust, remaining largely unaffected by extended condensation time or by changes in cell and nuclear size. In contrast, and as expected, changes in condensin levels exert a stronger influence on chromosome shape, similarly affecting both width and length. The conclusions are well supported by the data. Although the results do not directly reveal a mechanism of chromosome condensation, they provide valuable parameters that will inform further development of models of chromosome folding.

****Major Comments****

While chromosome width measurements are well described in the Methods section, the determination of chromosome arm length is less clear (even in Fig. S2B). How do the authors define the ends of a chromosome arm, particularly the centromeric end, which is embedded in a clustered centromere region? How are the size differences among the three chromosomes and their six arms accounted for? Since the observed differences are relatively small, these methodological points are critical.

Quantifying protein levels by western blotting is notoriously challenging. Could the authors complement these measurements with proteomics? More importantly, please clarify whether the reported values represent condensin numbers per cell or per nuclear/cellular volume. It would be useful to provide estimates for both, across all tested conditions, to assess which measure correlates better with the observed changes.

Chromosome arm length shows little variation across conditions; however, there appear to be notable differences between experiments. For instance, in Fig. 3 chromosomes appear longer than in other datasets. Could the authors provide an explanation for this variability?

****Minor Points****

The statement that chromosomes are "out-of-equilibrium structures" is confusing, as chromosome shaping is well established to be an energy-dependent process. Please clarify. The phrase "on the way to a steady state that is likely dictated by principles of polymer physics" requires further explanation.

Can the authors speculate whether their observations support a mechanism involving loop capture, loop extrusion, or whether they are consistent with both? It would be helpful to have the main models briefly introduced for the reader to put the work into context.

Significance

This study carefully quantifies chromosome shape parameters under mild cellular perturbations in *S. pombe*. The results indicate that chromosome shape is largely determined intrinsically, with minimal dependence on external physical factors, consistent with current understanding.

These findings will aid experts in chromosome biology in refining their models.

Reviewer expertise: condensin biology.

Reviewer 2

Evidence, reproducibility and clarity

Summary:

Provide a short summary of the findings and key conclusions (including methodology and model system(s) where appropriate).

This study investigates how the dimensions (length and width) of mitotic chromosomes are determined. The authors used *Schizosaccharomyces pombe* (fission yeast) as a model organism and combined super-resolution microscopy with semi-automated quantitative analysis and genetic manipulations. Previous studies have proposed that cell volume, nuclear volume, and the duration of chromosome condensation influence the size of mitotic chromosomes. However, in the present work, the authors found no correlation between chromosome dimensions and either cell or nuclear size, nor did they detect any effect of condensation duration. Instead, the analyses revealed a clear correlation between the amount of condensin complex and chromosome dimensions: reducing condensin levels led to an increase in chromosome width and length, whereas increasing condensin reduced them. These results suggest that the intrinsic amount of condensin, rather than extrinsic cellular or temporal factors, plays a key role in determining mitotic chromosome dimensions in fission yeast.

Major comments:

- Are the key conclusions convincing?

1. Use of arrested cells for chromosome dimension measurements:

All measurements were performed in synchronized populations-metaphase-arrested cells for chromosome width and G2-arrested cells for chromosome length. The authors should provide evidence that the chromosome dimensions measured under these arrest conditions are comparable to those in normally cycling cells, or discuss the rationale for assuming that the arrested conditions do not distort chromosome morphology.

2. Uncertainty in measuring chromosome length:

It is unclear whether chromosome length was accurately traced in all cases.

In Figure S2B, the thicker chromosomal segment appears traceable, but the thinner or faint structures are difficult to follow precisely, raising concerns about endpoint determination. To better evaluate the validity of the tracing approach, the authors should include examples showing the tracing of all six chromatids in Fig. S2B.

In addition, they should discuss how reliably the chromosome endpoints-especially those near the spindle poles-can be determined and how this affects the accuracy of the measured lengths.

3. Uncertainty in width measurements:

In the current data, although statistical tests yield significant p-values, the plotted distributions appear to overlap extensively, suggesting that the differences are subtle at best. Therefore, it remains unclear whether these measurements can robustly detect (or exclude) effects of cell size, nuclear size, or condensation time. The authors should more clearly justify the accuracy and sensitivity of their width measurements and explain why the chosen statistical tests are

appropriate given the variability and potential measurement uncertainty.

4. Interpretation of the condensation time experiment (Figure 4B)

In the analysis of condensation time, even the shortest treatment corresponds to approximately 3 hours of mitotic arrest, which may already represent a saturated state of chromosome condensation. To more directly assess the possible time dependence, it would therefore be most informative to include a zero-time (no-drug) condition as a reference, as noted above in comment 1. Such data would help determine whether chromosome dimensions are indeed unaffected by condensation duration or whether the effect is simply saturated under all tested conditions.

- Should the authors qualify some of their claims as preliminary or speculative, or remove them altogether?

5. Unless the issues raised above are adequately addressed, the corresponding conclusions should be toned down.

- Would additional experiments be essential to support the claims of the paper? Request additional experiments only where necessary for the paper as it is, and do not ask authors to open new lines of experimentation.

6. Need for validation of chromosome dimension measurements in non-arrested cells

As related to concerns 1 and 4 above, it would be important to measure or at least estimate the length and width of mitotic chromosomes in non-synchronized (non-arrested) cells. Such data are essential to demonstrate that the values obtained under arrest conditions are representative of normal physiology.

7. Since the measurement of chromosome dimensions (length and width) is the central theme of the study, an independent imaging or quantification method-would substantially strengthen the credibility of the findings.

- Are the suggested experiments realistic in terms of time and resources? It would help if you could add an estimated cost and time investment for substantial experiments.

7. I am not familiar with which experimental approaches are most effective in *Schizosaccharomyces pombe*. However, is it possible to perform chromosome-spread preparations to quantify chromosome dimensions? It would be helpful if the authors could consider whether established methods used in other organisms could be adapted for fission yeast. If no realistic alternatives exist beyond the current approach, the authors should at least discuss why conventional methods-such as chromosome spreading-are technically or biologically difficult to apply in this system.

- Are the data and the methods presented in such a way that they can be reproduced?

8. yes

- Are the experiments adequately replicated and statistical analysis adequate?

9. As far as I can tell, the statistical analyses appear to have been performed appropriately.

However, since I am not a statistician, I may not be in a position to fully assess their validity.

When looking at the plots, the differences between conditions are not visually obvious, and the interpretation therefore relies heavily on the statistical testing. Given the large sample sizes, the reported p-values are not particularly small, raising doubts as to whether the observed differences are truly meaningful. The authors should clarify how they interpret these results and whether the statistical outcomes indeed support the stated conclusions.

****Minor comments:****

- Specific experimental issues that are easily addressable.

10. In the analysis of condensation time, it may be more informative to evaluate changes in chromosome length rather than width. Length measurements appear to show clearer differences and, importantly, do not require metaphase arrest. Assessing condensation kinetics based on chromosome length would therefore provide a more direct and less perturbative evaluation.

- Are prior studies referenced appropriately?

11. yes

- Are the text and figures clear and accurate?

12. Overall, the text and figures are clear and well presented. However, as noted above, it seems technically challenging to measure mitotic chromosome dimensions accurately in *Schizosaccharomyces pombe*. Given these inherent difficulties, the rationale for choosing fission yeast as the model organism for this type of analysis is not entirely clear. The authors should explicitly explain why *S. pombe* is advantageous or appropriate for studying chromosome dimensions in this context.

- Do you have suggestions that would help the authors improve the presentation of their data and conclusions?

13. I suggest the authors to include schematic timelines in the figures to indicate when drug addition/removal and image acquisition occurred in each experiment. This would help readers easily understand the time course and treatment conditions associated with each dataset.

****Referees cross-commenting****

I have read the other two referee reports and found them reasonable and well balanced. Some of their comments overlap with my own, and even those that do not appear well justified and worth addressing by the authors.

Although it may be difficult to perform all the requested experiments, I strongly encourage the authors to discuss the methodological limitations and how these might affect the interpretation of their conclusions. Providing a transparent and balanced discussion of these aspects would substantially strengthen the manuscript.

Regarding the novelty of the study, I think the concern raised by another reviewer is valid. At the same time, if the robustness of the data can be further strengthened and the authors provide a thorough discussion of the relevant prior work while positioning their findings fairly within that context, the paper would hold value for the field.

Significance

- Describe the nature and significance of the advance (e.g. conceptual, technical, clinical) for the field.

The finding that condensin influences the dimensions (width and length) of mitotic chromosomes is conceptually not entirely surprising, but the authors' demonstration of this relationship through direct in vivo manipulation is valuable and represents a meaningful technical contribution to the field.

However, while the study also concludes that chromosome dimensions are independent of cell size, nuclear size, and condensation time, these negative results should be interpreted with caution. Given the uncertainties in measurement accuracy discussed above, the data may not be sufficient to definitively exclude such influences. The authors are encouraged to moderate these claims accordingly.

- Place the work in the context of the existing literature (provide references, where appropriate).

As far as I can tell, the Introduction accurately and appropriately summarizes the relevant literature. The authors provide a clear overview of previous findings on chromosome dimensions (length and width) across different species and the molecular factors implicated in these processes.

Importantly, while various studies have described correlations between chromosome morphology and specific proteins, there appear to be very few examples in which the effects of condensin on chromosome dimensions have been directly manipulated and quantified in vivo. Therefore, this study makes a meaningful contribution by providing direct experimental evidence on this point.

- State what audience might be interested in and influenced by the reported findings.

The work will be of interest to researchers studying chromosome architecture and organization. It may also attract attention from those interested in cellular- and nuclear-scale size control mechanisms, even beyond the field of chromosome biology.

- Define your field of expertise with a few keywords to help the authors contextualize your point of view. Indicate if there are any parts of the paper that you do not have sufficient expertise to evaluate.

I am a cell biologist with a particular interest and research focus in size control mechanisms. I do not have direct experimental experience with *Schizosaccharomyces pombe*. While I have general knowledge of statistical analysis, I am not an expert in the specific statistical approaches required to rigorously evaluate subtle quantitative differences such as those analyzed in this study.

Reviewer 3**Evidence, reproducibility and clarity******Summary****

Wu and colleagues quantified chromosome area, arm width, and arm length in fission yeast by measuring DAPI-stained chromatin in cells released into mitosis after varying durations of G2-phase arrest. During prolonged G2 arrest, chromatin area expanded continuously, in proportion to increases in nuclear and cell size. Upon entry into mitosis, chromosomes re-compacted to a similar extent, restoring arm width, but not arm length, regardless of the length of G2 arrest.

In mutant strains with smaller cell size, mitotic chromosomes likewise showed no major differences in occupied area or arm width but exhibited somewhat increased arm lengths. Western blot analysis indicated that the concentration of the condensin subunit Cut14 (SMC2) was reduced both, in larger cells following prolonged G2 arrest and in smaller mutant cells. Gradual depletion of Cut14 led to a corresponding increase in metaphase chromosome arm width and, when Cut14 levels were reduced to roughly half, an increase in anaphase chromosome length. Conversely, co-overexpression of the five condensin subunits caused a slight decrease in metaphase arm width and anaphase chromosome length.

The authors conclude that mitotic chromosome compaction depends on the cellular level of condensin, but is independent of cell or nuclear size.

****Major comments****

1. The authors conclude that reduced condensin levels in larger or smaller cells limit chromosome arm shortening (p. 8), based on Western blot analyses of a single condensin subunit (Figs. 1D and 2D). However, this interpretation would only be valid if Cut14 were the rate-limiting component of the five-subunit condensin complex; an assumption that seems unlikely.

Furthermore, even if Cut14 were rate-limiting, the manuscript does not demonstrate that a 10-30 % reduction in total cellular Cut14 levels leads to a corresponding decrease in condensin abundance on chromosomes, as claimed in the abstract ("Instead, levels of the chromosomal condensin complex..."). In the Discussion, the authors acknowledge that quantitative assessment of chromosome-bound condensin is "experimentally challenging." Nonetheless, established methods such as spike-in ChIP-seq offer a feasible and quantitative approach to measure chromatin-associated SMC protein levels genome-wide (Hu et al., *Nucleic Acids Res.*, 2015).

2. The authors use "chromatin area" as a measure of chromosome compaction. However, it is unclear from the Methods section whether this area was derived from a maximum-intensity projection of the DAPI signal or from a single z-slice. In either case, quantifying chromatin volume would seem more appropriate for assessing chromosome compaction. Additionally, the use of a 500 nm z-spacing raises concerns, given that the fission yeast nucleus has a diameter of only about 1 μm .

3. Another concern relates to the measurement of chromosome length in anaphase cells. It is unclear how the authors accurately identified the centromere position of individual chromosomes within clustered chromatids. The Methods section lacks a description of how chromosome arm lengths were determined.

Previous studies have employed fluorescently labeled reporter arrays to precisely measure chromosome lengths in yeast species, including fission yeast (e.g., Sakuno et al., *Nature* 2009; Petrova et al., *Mol. Cell Biol.* 2013). Given that the main conclusions of this work rely on very small differences in chromosome length (see next comment), it is essential that the authors validate their findings using such established, high-resolution approaches available for their model system. Moreover, the authors should consider that apparent differences in anaphase chromosome arm length may reflect stretching by microtubule pulling forces. The use of reporter arrays would enable measurements in metaphase-arrested cells, mitigating this concern.

4. Although many of the reported differences reach statistical significance, the corresponding

effect sizes are often small. For instance, the measured chromosome arm lengths differ by only 0.1-0.15 μm , less than 10 % of the total arm length (Figs. 1C, 2C, and 3C). The reduction observed upon condensin overexpression is even smaller, at approximately 0.05 μm (Fig. 4C). Notably, the authors describe this minor difference as "significantly shorter" ($p = 0.03$), yet refer to the reduction in chromosome width in the *ppa2* mutant as "marginally reduced" (Fig. 2B), despite a comparable level of statistical significance ($p = 0.02$).

5. Assessing the statistical significance of Western blot signals (Figs. 1D, 2D, and 3C) based on a sample size of $n = 3$ is questionable, particularly if signal linearity cannot be ensured when using enzyme-linked secondary antibodies (e.g., HRP), as signal saturation may occur. The Methods section does not specify what steps were taken by the authors to verify linearity or to ensure accurate signal quantification.

As an alternative approach, the authors could consider fluorescently tagging condensin subunits and comparing total fluorescence signals between strains to obtain a quantitative measure of protein levels.

****Minor comments****

1. It is unclear why the authors arrested cells and depleted Cdc14 by auxin addition for 5 hours to measure metaphase chromosome arm widths (Fig. 3B), but only for 3 hours (plus the time until anaphase) to measure anaphase chromosome lengths (Fig. 3C). This discrepancy likely accounts for the observed difference in Cdc14 depletion levels- approximately 10 % remaining at 500 μM IAA in the former experiment versus about 50 % in the latter. To enable a more meaningful comparison between the two conditions, the authors could consider shortening the Slp1 arrest to 3.5 hours, which should allow complete metaphase arrest while achieving condensin depletion levels comparable to those in the anaphase release experiment.

2. The authors contend that the observed shortening of chromosome length upon condensin overexpression supports a loop capture model rather than a loop extrusion model. However, under a loop extrusion mechanism, an excess of condensin would be expected to produce smaller loops and, consequently, longer chromosome arms only if colliding condensin complexes were unable to bypass one another, which does not appear to be the case (Kim et al., Nature 2020).

3. The authors propose that alterations in transcriptional regulation may underlie the reduced cell and nuclear size observed upon condensin depletion, citing their 2021 publication. However, they do not acknowledge previous evidence demonstrating that condensin does not play a direct role in transcriptome maintenance in fission yeast (Hocquet et al., eLife, 2018). This apparent discrepancy should be explicitly discussed.

Significance

The question of which factors shape mitotic chromosomes has long been a central issue in cell biology. Foundational insights into this problem, particularly the pivotal role of condensin in organizing mitotic chromosomes, have come from experiments using *Xenopus* egg extracts. However, this *in vitro* system does not allow for assessing the influence of cell or nuclear size on mitotic chromosome architecture. The demonstration that mitotic chromosomes do not scale with cell or nuclear size in fission yeast therefore represents a notable contribution.

Nonetheless, the observation that chromosome length is determined by condensin is largely consistent with previous findings and thus not entirely unexpected. Previous studies have shown that anaphase chromosome length increases in fission yeast condensin mutants (Petrova et al., Mol Cell Biol 2013), including strains in which expression of the condensin subunit Cnd1 is reduced by half (Schiklenk et al., J Cell Biol 2018). Surprisingly, none of these studies are cited in the manuscript. Instead, the majority of references point primarily to the authors' own previous work.

In conclusion, while the study offers interesting observations, its overall novelty appears limited, and several experimental concerns remain unresolved. Substantial additional experimentation would be needed to fully support the conclusions, and these revisions may extend beyond what is feasible for a standard revision.

Manuscript number: RC -2025-03217

Corresponding author(s): Frank Uhlmann

1. General Statements [optional]

N/A

2. Description of the planned revisions

We would like to thank the three reviewers for their constructive feedback on our manuscript, which we propose to use in the following ways to improve and strengthen our study.

Reviewer #1 (Evidence, reproducibility and clarity (Required)):

Summary

*In this manuscript, Wu et al investigate the shape of mitotic chromosomes under normal conditions and after altering cell cycle parameters and condensin levels. By quantitatively measuring chromosome width and length, they show that chromosome shape in *S. pombe* is remarkably robust, remaining largely unaffected by extended condensation time or by changes in cell and nuclear size. In contrast, and as expected, changes in condensin levels exert a stronger influence on chromosome shape, similarly affecting both width and length.*

The conclusions are well supported by the data. Although the results do not directly reveal a mechanism of chromosome condensation, they provide valuable parameters that will inform further development of models of chromosome folding.

We thank the reviewer for an accurate summary of our work.

Major Comments

While chromosome width measurements are well described in the Methods section, the determination of chromosome arm length is less clear (even in Fig. S2B). How do the authors define the ends of a chromosome arm, particularly the centromeric end, which is embedded in a clustered centromere region? How are the size differences among the three chromosomes and their six arms accounted for? Since the observed differences are relatively small, these methodological points are critical.

We will expand the methods section to better describe our chromosome arm length measurements. All clearly discernible arms are measured from the centre of the centromere cluster. The DNA stain does not distinguish between the six fission yeast chromosome arms, so the measurements are an aggregate of the lengths of all six arms.

Quantifying protein levels by western blotting is notoriously challenging. Could the authors complement these measurements with proteomics? More importantly, please clarify whether the reported values represent condensin numbers per cell or per nuclear/cellular volume. It would be useful to provide estimates for both, across all tested conditions, to assess which measure correlates better with the observed changes.

Western blotting is a suitable way to assess protein levels. We ensured that all quantified signals were in the linear detection range, and we performed all quantifications in three independent biological repeat experiments. Condensin levels were normalised to the level of housekeeping metabolic enzymes, whose concentration is known to remain constant in cells of increasing (or decreasing) sizes. The levels therefore reflect condensin concentration (i.e. 'condensin numbers per cellular volume').

Chromosome arm length shows little variation across conditions; however, there appear to be notable differences between experiments. For instance, in Fig. 3 chromosomes appear longer than in other datasets. Could the authors provide an explanation for this variability?

Chromosome arm lengths are comparable between most experiments, though lengths appear overall greater in the experiment shown in Figure 3. We do not currently know the reason for this apparent deviation. The experiment uses a different fission yeast strain background, which might have contributed to the difference. This said, the comparison shown in the Figure is internally controlled, i.e. the same strain is compared under two experimental conditions.

Minor Points

The statement that chromosomes are "out-of-equilibrium structures" is confusing, as chromosome shaping is well established to be an energy-dependent process. Please clarify.

We will replace this phrase with "structures that are out of steady state".

The phrase "on the way to a steady state that is likely dictated by principles of polymer physics" requires further explanation.

Further explanation will be provided.

Can the authors speculate whether their observations support a mechanism involving loop capture, loop extrusion, or whether they are consistent with both? It would be helpful to have the main models briefly introduced for the reader to put the work into context.

We will expand the discussion of how our observations inform chromosome formation models. The loop extrusion model entails that fewer condensins generate fewer but longer loops, i.e. chromosomes should become wider and shorter, contrary to our observation of wider and longer chromosomes. In a loop capture scenario, fewer condensins would result in both wider and longer chromosome, in line with our observations.

Reviewer #1 (Significance (Required)):

*This study carefully quantifies chromosome shape parameters under mild cellular perturbations in *S. pombe*. The results indicate that chromosome shape is largely determined intrinsically, with minimal dependence on external physical factors, consistent with current understanding.*

We note that the chromosome size literature did put forward control principles that depend on external physical factors (e.g. Kimura et al 2013, Zhou et al. 2023). Our study was specifically designed to explore such factors, using the setting of a simple and genetically controllable model organism.

These findings will aid experts in chromosome biology in refining their models.

Reviewer expertise: condensin biology.

Reviewer #2 (Evidence, reproducibility and clarity (Required)):**Summary:**

Provide a short summary of the findings and key conclusions (including methodology and model system(s) where appropriate).

*This study investigates how the dimensions (length and width) of mitotic chromosomes are determined. The authors used *Schizosaccharomyces pombe* (fission yeast) as a model organism and combined super-resolution microscopy with semi-automated quantitative analysis and genetic manipulations. Previous studies have proposed that cell volume, nuclear volume, and the duration of chromosome condensation influence the size of mitotic chromosomes. However, in the present work, the authors found no correlation between chromosome dimensions and either cell or nuclear size, nor did they detect any effect of condensation duration. Instead, the analyses revealed a clear correlation between the amount of condensin complex and chromosome dimensions: reducing condensin levels led to an increase in chromosome width and length, whereas increasing condensin reduced them. These results suggest that the intrinsic amount of condensin, rather than extrinsic cellular or temporal factors, plays a key role in determining mitotic chromosome dimensions in fission yeast.*

We thank the reviewer for an accurate summary of our work.

Major comments:

- Are the key conclusions convincing?

1. Use of arrested cells for chromosome dimension measurements:

All measurements were performed in synchronized populations-metaphase-arrested cells for chromosome width and G2-arrested cells for chromosome length. The authors should provide

evidence that the chromosome dimensions measured under these arrest conditions are comparable to those in normally cycling cells, or discuss the rationale for assuming that the arrested conditions do not distort chromosome morphology.

Mitosis covers only a short period of the cell growth and division cycle, and mitotic cells are rare without cell synchronisation. Nevertheless, as part of our revision experiments, we will turn to asynchronously growing cell populations and will aim to capture chromosome dimensions in the naturally occurring fraction of mitotic cells.

2. Uncertainty in measuring chromosome length:

It is unclear whether chromosome length was accurately traced in all cases.

In Figure S2B, the thicker chromosomal segment appears traceable, but the thinner or faint structures are difficult to follow precisely, raising concerns about endpoint determination. To better evaluate the validity of the tracing approach, the authors should include examples showing the tracing of all six chromatids in Fig. S2B.

In addition, they should discuss how reliably the chromosome endpoints-especially those near the spindle poles-can be determined and how this affects the accuracy of the measured lengths.

We will expand the methods section to better describe our chromosome arm length measurements. All discernible arms are measured from the centre of the centromere cluster. However, if the chromosome outline is not unambiguous, the arm is excluded from the analysis. Typically, therefore, not all six chromosome arms are measured.

3. Uncertainty in width measurements:

In the current data, although statistical tests yield significant p-values, the plotted distributions appear to overlap extensively, suggesting that the differences are subtle at best.

Therefore, it remains unclear whether these measurements can robustly detect (or exclude) effects of cell size, nuclear size, or condensation time. The authors should more clearly justify the accuracy and sensitivity of their width measurements and explain why the chosen statistical tests are appropriate given the variability and potential measurement uncertainty.

The reviewer raises an important limitation of approaches to characterise chromosome dimensions. Overlapping width and length distributions were encountered in previous studies that measured chromosome dimensions in both yeasts and vertebrates (e.g. Kimura et al 2013, Kakui et al. 2022, Zhou et al. 2023). Despite the overlap, the measurements and their statistical analysis can identify parameters that lead to changes in the size distributions.

4. Interpretation of the condensation time experiment (Figure 4B)

In the analysis of condensation time, even the shortest treatment corresponds to approximately 3 hours of mitotic arrest, which may already represent a saturated state of chromosome condensation. To more directly assess the possible time dependence, it would therefore be most informative to include a zero-time (no-drug) condition as a reference, as noted above in comment 1. Such data would help determine whether chromosome dimensions are indeed unaffected by condensation duration or whether the effect is simply saturated under all tested conditions.

This point relates to comment 1. During our revisions, we will turn to asynchronously growing cell populations and capture chromosome dimensions in the naturally occurring fraction of mitotic cells.

- Should the authors qualify some of their claims as preliminary or speculative, or remove them altogether?

5. Unless the issues raised above are adequately addressed, the corresponding conclusions should be toned down.

We will do our best to address the issues raised.

- Would additional experiments be essential to support the claims of the paper? Request additional experiments only where necessary for the paper as it is, and do not ask authors to open new lines of experimentation.

6. Need for validation of chromosome dimension measurements in non-arrested cells

As related to concerns 1 and 4 above, it would be important to measure or at least estimate the length and width of mitotic chromosomes in non-synchronized (non-arrested) cells. Such data are

essential to demonstrate that the values obtained under arrest conditions are representative of normal physiology.

See above, points 1 and 4

7. Since the measurement of chromosome dimensions (length and width) is the central theme of the study, an independent imaging or quantification method-would substantially strengthen the credibility of the findings.

- Are the suggested experiments realistic in terms of time and resources? It would help if you could add an estimated cost and time investment for substantial experiments.

7. I am not familiar with which experimental approaches are most effective in Schizosaccharomyces pombe. However, is it possible to perform chromosome-spread preparations to quantify chromosome dimensions? It would be helpful if the authors could consider whether established methods used in other organisms could be adapted for fission yeast. If no realistic alternatives exist beyond the current approach, the authors should at least discuss why conventional methods-such as chromosome spreading-are technically or biologically difficult to apply in this system.

As an independent approach, as suggested by the reviewer, we have indeed performed chromosome-spread preparations, using paraformaldehyde as an alternative fixative. We will include some of the results obtained in a new supplemental figure. Measurements of spread chromosomes yield comparable numbers to those obtained from our whole mount samples. However, we noticed slide-to-slide variation when multiple slides were prepared from the same sample. The chromosome spreading process appeared to mildly distort the dimensions of fission yeast chromosome. To minimise distortion, we decided to conduct most routine measurements in fixed and stained whole mount samples.

- Are the data and the methods presented in such a way that they can be reproduced?

8. yes

- Are the experiments adequately replicated and statistical analysis adequate?

9. As far as I can tell, the statistical analyses appear to have been performed appropriately. However, since I am not a statistician, I may not be in a position to fully assess their validity. When looking at the plots, the differences between conditions are not visually obvious, and the interpretation therefore relies heavily on the statistical testing. Given the large sample sizes, the reported p-values are not particularly small, raising doubts as to whether the observed differences are truly meaningful. The authors should clarify how they interpret these results and whether the statistical outcomes indeed support the stated conclusions.

We will revisit all quantitative comparisons and review their reliability.

Minor comments:

- Specific experimental issues that are easily addressable.

10. In the analysis of condensation time, it may be more informative to evaluate changes in chromosome length rather than width. Length measurements appear to show clearer differences and, importantly, do not require metaphase arrest. Assessing condensation kinetics based on chromosome length would therefore provide a more direct and less perturbative evaluation.

The reviewer makes an excellent suggestion, to measure anaphase chromosome lengths after increasing times of mitotic arrest. We have attempted to perform this experiment. Mitotic arrest is achieved by thiamine-induced repression of the Slp1 cell cycle regulator under control of the *nmt41* promoter. To resume mitotic progression, we must re-activate the *nmt41* promoter. However, while *nmt41* promoter repression by thiamine addition is fast, re-activation after thiamine removal

is slow (14 - 20 hours). This slow response unfortunately precluded us from successfully releasing mitotically arrested cells for monitoring chromosomes during anaphase.

- Are prior studies referenced appropriately?

11. yes

- Are the text and figures clear and accurate?

12. Overall, the text and figures are clear and well presented. However, as noted above, it seems technically challenging to measure mitotic chromosome dimensions accurately in *Schizosaccharomyces pombe*. Given these inherent difficulties, the rationale for choosing fission yeast as the model organism for this type of analysis is not entirely clear. The authors should explicitly explain why *S. pombe* is advantageous or appropriate for studying chromosome dimensions in this context.

Yeast models offer exquisite genetic control over cellular variables, like cell and nuclear size and cell cycle stage. Amongst the yeast models, fission yeast has three, relatively large chromosomes. These considerations motivated our model organism choice. At the same time, we concede that yeast mitotic chromosomes are harder to image than those of higher eukaryotes.

- Do you have suggestions that would help the authors improve the presentation of their data and conclusions?

13. I suggest the authors to include schematic timelines in the figures to indicate when drug addition/removal and image acquisition occurred in each experiment. This would help readers easily understand the time course and treatment conditions associated with each dataset.

Thank you for this suggestion, which we will implement.

****Referees cross-commenting****

I have read the other two referee reports and found them reasonable and well balanced. Some of their comments overlap with my own, and even those that do not appear well justified and worth addressing by the authors.

Although it may be difficult to perform all the requested experiments, I strongly encourage the authors to discuss the methodological limitations and how these might affect the interpretation of their conclusions. Providing a transparent and balanced discussion of these aspects would substantially strengthen the manuscript.

Regarding the novelty of the study, I think the concern raised by another reviewer is valid. At the same time, if the robustness of the data can be further strengthened and the authors provide a thorough discussion of the relevant prior work while positioning their findings fairly within that context, the paper would hold value for the field.

We will try our best to improve the robustness of the presented data, and we will revise the manuscript text to provide a transparent and balanced discussion.

Reviewer #2 (Significance (Required)):

- Describe the nature and significance of the advance (e.g. conceptual, technical, clinical) for the field.

The finding that condensin influences the dimensions (width and length) of mitotic chromosomes is conceptually not entirely surprising, but the authors' demonstration of this relationship through direct in vivo manipulation is valuable and represents a meaningful technical contribution to the field.

However, while the study also concludes that chromosome dimensions are independent of cell size, nuclear size, and condensation time, these negative results should be interpreted with caution. Given the uncertainties in measurement accuracy discussed above, the data may not be sufficient

to definitively exclude such influences. The authors are encouraged to moderate these claims accordingly.

- Place the work in the context of the existing literature (provide references, where appropriate).

As far as I can tell, the Introduction accurately and appropriately summarizes the relevant literature. The authors provide a clear overview of previous findings on chromosome dimensions (length and width) across different species and the molecular factors implicated in these processes.

Importantly, while various studies have described correlations between chromosome morphology and specific proteins, there appear to be very few examples in which the effects of condensin on chromosome dimensions have been directly manipulated and quantified *in vivo*. Therefore, this study makes a meaningful contribution by providing direct experimental evidence on this point.

- State what audience might be interested in and influenced by the reported findings.

The work will be of interest to researchers studying chromosome architecture and organization. It may also attract attention from those interested in cellular- and nuclear-scale size control mechanisms, even beyond the field of chromosome biology.

- Define your field of expertise with a few keywords to help the authors contextualize your point of view. Indicate if there are any parts of the paper that you do not have sufficient expertise to evaluate.

I am a cell biologist with a particular interest and research focus in size control mechanisms. I do not have direct experimental experience with *Schizosaccharomyces pombe*. While I have general knowledge of statistical analysis, I am not an expert in the specific statistical approaches required to rigorously evaluate subtle quantitative differences such as those analyzed in this study.

Reviewer #3 (Evidence, reproducibility and clarity (Required)):

Summary

Wu and colleagues quantified chromosome area, arm width, and arm length in fission yeast by measuring DAPI-stained chromatin in cells released into mitosis after varying durations of G2-phase arrest. During prolonged G2 arrest, chromatin area expanded continuously, in proportion to increases in nuclear and cell size. Upon entry into mitosis, chromosomes re-compacted to a similar extent, restoring arm width, but not arm length, regardless of the length of G2 arrest.

In mutant strains with smaller cell size, mitotic chromosomes likewise showed no major differences in occupied area or arm width but exhibited somewhat increased arm lengths. Western blot analysis indicated that the concentration of the condensin subunit Cut14 (SMC2) was reduced both, in larger cells following prolonged G2 arrest and in smaller mutant cells. Gradual depletion of Cut14 led to a corresponding increase in metaphase chromosome arm width and, when Cut14 levels were reduced to roughly half, an increase in anaphase chromosome length. Conversely, co-overexpression of the five condensin subunits caused a slight decrease in metaphase arm width and anaphase chromosome length.

The authors conclude that mitotic chromosome compaction depends on the cellular level of condensin, but is independent of cell or nuclear size.

We thank the reviewer for an accurate summary of our work.

Major comments

1. The authors conclude that reduced condensin levels in larger or smaller cells limit chromosome arm shortening (p. 8), based on Western blot analyses of a single condensin subunit (Figs. 1D and

2D). However, this interpretation would only be valid if Cut14 were the rate-limiting component of the five-subunit condensin complex; an assumption that seems unlikely.

Condensin is a stoichiometric five-subunit protein complex, and each of its five subunits is essential for its function. Therefore, reducing the levels of any one of the five components reduces the level of functional condensin complexes.

Furthermore, even if Cut14 were rate-limiting, the manuscript does not demonstrate that a 10-30 % reduction in total cellular Cut14 levels leads to a corresponding decrease in condensin abundance on chromosomes, as claimed in the abstract ("Instead, levels of the chromosomal condensin complex..."). In the Discussion, the authors acknowledge that quantitative assessment of chromosome-bound condensin is "experimentally challenging." Nonetheless, established methods such as spike-in ChIP-seq offer a feasible and quantitative approach to measure chromatin-associated SMC protein levels genome-wide (Hu et al., *Nucleic Acids Res.*, 2015).

Unlike cohesin, studied by Hu et al. 2015, condensin is known to be a much trickier target for (quantitative) ChIP analyses. Establishing a robust condensin ChIP approach, while desirable, will go beyond the possible scope of our revisions.

2. The authors use "chromatin area" as a measure of chromosome compaction. However, it is unclear from the Methods section whether this area was derived from a maximum-intensity projection of the DAPI signal or from a single z-slice. In either case, quantifying chromatin volume would seem more appropriate for assessing chromosome compaction. Additionally, the use of a 500 nm z-spacing raises concerns, given that the fission yeast nucleus has a diameter of only about 1 μm .

Indeed, we used maximum intensity projections to derive the input for the ilastik tool to segment the chromatin-occupied area. We will amend the methods section to clarify our approach. 3D segmentation would be much more demanding, and likely less accurate.

3. Another concern relates to the measurement of chromosome length in anaphase cells. It is unclear how the authors accurately identified the centromere position of individual chromosomes within clustered chromatids. The Methods section lacks a description of how chromosome arm lengths were determined.

We will expand the methods section to better describe our chromosome arm length measurements. All clearly discernible arms are measured from the centre of the centromere cluster.

Previous studies have employed fluorescently labeled reporter arrays to precisely measure chromosome lengths in yeast species, including fission yeast (e.g., Sakuno et al., *Nature* 2009; Petrova et al., *Mol. Cell Biol.* 2013). Given that the main conclusions of this work rely on very small differences in chromosome length (see next comment), it is essential that the authors validate their findings using such established, high-resolution approaches available for their model system. Moreover, the authors should consider that apparent differences in anaphase chromosome arm length may reflect stretching by microtubule pulling forces. The use of reporter arrays would enable measurements in metaphase-arrested cells, mitigating this concern.

We contemplated using one of the previously developed reporter array systems, but decided against their use for the following reasons. The reporters mark two loci at shorter distance than chromosome arm length, so any small chromosome arm length changes would be manifest in even smaller array distances changes. Furthermore, while array distances can be measured in metaphase, the chromosome path between the two loci, and with it the chromosome contour length, remain unknown.

4. Although many of the reported differences reach statistical significance, the corresponding effect sizes are often small. For instance, the measured chromosome arm lengths differ by only 0.1-0.15 μm , less than 10 % of the total arm length (Figs. 1C, 2C, and 3C). The reduction observed upon condensin overexpression is even smaller, at approximately 0.05 μm (Fig. 4C). Notably, the authors describe this minor difference as "significantly shorter" ($p = 0.03$), yet refer to the reduction in chromosome width in the *ppa2* mutant as "marginally reduced" (Fig. 2B), despite a comparable level of statistical significance ($p = 0.02$).

We will look back at our descriptions to ensure that they all accurately reflect the measurements. Indeed, the finding that chromosome size changes are small, despite vast cell and nuclear size changes, is one of the key conclusions from our study.

5. Assessing the statistical significance of Western blot signals (Figs. 1D, 2D, and 3C) based on a sample size of $n = 3$ is questionable, particularly if signal linearity cannot be ensured when using enzyme-linked secondary antibodies (e.g., HRP), as signal saturation may occur. The Methods section does not specify what steps were taken by the authors to verify linearity or to ensure accurate signal quantification.

Western blotting is a suitable method to quantitatively assess protein levels. We ensured that all quantified signals were in the linear detection range, and we performed all quantifications in three independent biological repeat experiments. We will add further detail to the methods section on how band intensities were quantified.

As an alternative approach, the authors could consider fluorescently tagging condensin subunits and comparing total fluorescence signals between strains to obtain a quantitative measure of protein levels.

Fluorescence signal quantification is an alternative method to measure protein levels, which comes with its own confounding factors relating to signal quenching, bleaching and background correction.

Minor comments

1. It is unclear why the authors arrested cells and depleted Cdc14 by auxin addition for 5 hours to measure metaphase chromosome arm widths (Fig. 3B), but only for 3 hours (plus the time until anaphase) to measure anaphase chromosome lengths (Fig. 3C). This discrepancy likely accounts for the observed difference in Cdc14 depletion levels—approximately 10 % remaining at 500 μ M IAA in the former experiment versus about 50 % in the latter. To enable a more meaningful comparison between the two conditions, the authors could consider shortening the Slp1 arrest to 3.5 hours, which should allow complete metaphase arrest while achieving condensin depletion levels comparable to those in the anaphase release experiment.

5 hours depletion were used in Figure 3B to reach a wide dynamic depletion range, and the reviewer is correct that similar depletion levels were inaccessible within only 3 hours. The 3-hour limit in the experiment shown in Figure 3C was imposed by the practicalities of synchronous anaphase release in cell of increasing size.

2. The authors contend that the observed shortening of chromosome length upon condensin overexpression supports a loop capture model rather than a loop extrusion model. However, under a loop extrusion mechanism, an excess of condensin would be expected to produce smaller loops and, consequently, longer chromosome arms only if colliding condensin complexes were unable to bypass one another, which does not appear to be the case (Kim et al., Nature 2020).

Whether bypass occurs if condensins were to extrude chromatin loops *in vivo* is not yet known and is currently a topic of debate in the field. Irrespective of bypass, the loop extrusion model suggests that fewer condensins generate fewer and longer loops, resulting in wider and shorter chromosomes. This prediction is contrary to our observation that chromosomes become wider and longer.

3. The authors propose that alterations in transcriptional regulation may underlie the reduced cell and nuclear size observed upon condensin depletion, citing their 2021 publication. However, they do not acknowledge previous evidence demonstrating that condensin does not play a direct role in transcriptome maintenance in fission yeast (Hocquet et al., eLife, 2018). This apparent discrepancy should be explicitly discussed.

We thank the reviewer for pointing us to the Hocquet et al. paper. These authors document gene expression changes that are due to cell division errors in the absence of condensin. They do not, however, study gene expression changes in the absence of cell divisions (apart from 3 ‘cell division-affected’ genes). We will aim to include a discussion of the Hocquet et al. study in our revised manuscript.

Reviewer #3 (Significance (Required)):

*The question of which factors shape mitotic chromosomes has long been a central issue in cell biology. Foundational insights into this problem, particularly the pivotal role of condensin in organizing mitotic chromosomes, have come from experiments using *Xenopus* egg extracts. However, this *in vitro* system does not allow for assessing the influence of cell or nuclear size on*

mitotic chromosome architecture. The demonstration that mitotic chromosomes do not scale with cell or nuclear size in fission yeast therefore represents a notable contribution.

Nonetheless, the observation that chromosome length is determined by condensin is largely consistent with previous findings and thus not entirely unexpected. Previous studies have shown that anaphase chromosome length increases in fission yeast condensin mutants (Petrova et al., Mol Cell Biol 2013), including strains in which expression of the condensin subunit Cnd1 is reduced by half (Schiklenk et al., J Cell Biol 2018). Surprisingly, none of these studies are cited in the manuscript. Instead, the majority of references point primarily to the authors' own previous work.

Apologies for overlooking these additional, relevant, papers from the Haering lab. These papers confirm, and quantitatively underscore, what was previously established by work in the Yanagida lab (that we referred to). We will refer to these additional studies in our revised manuscript.

In conclusion, while the study offers interesting observations, its overall novelty appears limited, and several experimental concerns remain unresolved. Substantial additional experimentation would be needed to fully support the conclusions, and these revisions may extend beyond what is feasible for a standard revision.

3. Description of the revisions that have already been incorporated in the transferred manuscript

We enclose our original submitted manuscript, and we aim to make the indicated revisions by the time of submitting the revised manuscript.

4. Description of analyses that authors prefer not to carry out

N/A

Original submission

First decision letter

MS ID#: jcs.264569

MS TITLE: Investigations into fission yeast chromosome size determinants

AUTHORS: Pei-Shang Wu; Todd Fallesen; Frank Uhlmann

ARTICLE TYPE: Review Commons Transfer

Dear Dr Uhlmann,

Many thanks for transferring your manuscript to Journal of Cell Science from Review Commons. I have now had the chance to review your documents, and we would like to invite you to revise your manuscript according to your revision plan. Once we receive the revised version, we may seek further input from the Review Commons referees. If you have any questions about this process, please do get in touch.

First revision

Author response to reviewers' comments

We would like to thank the three expert reviewers for their constructive feedback on our manuscript, which allowed us to improve and strengthen our study. We hope that, with these revisions, the reviewers will support publication of our manuscript in the *Journal of Cell Science*.

Reviewer #1 (Evidence, reproducibility and clarity (Required)):

Summary

*In this manuscript, Wu et al investigate the shape of mitotic chromosomes under normal conditions and after altering cell cycle parameters and condensin levels. By quantitatively measuring chromosome width and length, they show that chromosome shape in *S. pombe* is remarkably robust, remaining largely unaffected by extended condensation time or by changes in cell and nuclear size. In contrast, and as expected, changes in condensin levels exert a stronger influence on chromosome shape, similarly affecting both width and length.*

The conclusions are well supported by the data. Although the results do not directly reveal a mechanism of chromosome condensation, they provide valuable parameters that will inform further development of models of chromosome folding.

We thank the reviewer for an accurate appraisal of our work.

Major Comments

While chromosome width measurements are well described in the Methods section, the determination of chromosome arm length is less clear (even in Fig. S2B). How do the authors define the ends of a chromosome arm, particularly the centromeric end, which is embedded in a clustered centromere region? How are the size differences among the three chromosomes and their six arms accounted for? Since the observed differences are relatively small, these methodological points are critical.

We have added a methods section that describes our chromosome arm length measurements. All discernible arms, with clearly identifiable beginnings and ends, and pointing in the direction of anaphase separation, were included in the analysis. As the DNA stain does not distinguish between the six fission yeast chromosome arms, the measurements are an aggregate of the lengths of all six arms.

Quantifying protein levels by western blotting is notoriously challenging. Could the authors complement these measurements with proteomics? More importantly, please clarify whether the reported values represent condensin numbers per cell or per nuclear/cellular volume. It would be useful to provide estimates for both, across all tested conditions, to assess which measure correlates better with the observed changes.

Western blotting is a widely used, quantitative method to assess protein levels. We ensured that all quantified signals were in the linear detection range, and we performed all quantifications in three independent biological repeat experiments. Condensin levels were normalised to the level of housekeeping metabolic enzymes, whose concentration is known to remain constant in cells of different sizes. The measured condensin levels thus reflect condensin concentration (i.e. ‘condensin numbers per cellular volume’). Details of this approach have been included in an additional methods section.

Chromosome arm length shows little variation across conditions; however, there appear to be notable differences between experiments. For instance, in Fig. 3 chromosomes appear longer than in other datasets. Could the authors provide an explanation for this variability?

Chromosome arm lengths are comparable between most experiments, though the reviewer is right that lengths appear overall greater in the experiment shown in Figure 3. We do not currently know the reason for this apparent deviation. The experiment uses a different fission yeast strain background, which might have contributed to the difference. We emphasise that comparisons in each Figure are internally controlled, i.e. in Figure 3 one culture is split, and cells from the same population are then compared under the indicated experimental conditions. Images in each experiment were acquired side-by-side under identical imaging conditions.

Minor Points

The statement that chromosomes are "out-of-equilibrium structures" is confusing, as chromosome shaping is well established to be an energy-dependent process. Please clarify.

The reviewer is correct, and we have replaced “equilibrium” with “steady state”.

The phrase "on the way to a steady state that is likely dictated by principles of polymer physics" requires further explanation.

We have simplified this section of the introduction and removed this phrase.

Can the authors speculate whether their observations support a mechanism involving loop capture, loop extrusion, or whether they are consistent with both? It would be helpful to have the main models briefly introduced for the reader to put the work into context.

We had briefly compared the loop capture and loop extrusion models at the end of our discussion, and we have expanded this section to provide more background and clarity. In brief, the loop extrusion model predicts that a greater number of condensins generate more smaller loops, i.e. chromosomes should become thinner but also longer. This prediction is not met by our observation that chromosomes become progressively thinner and shorter as condensin dosage increases, an observation more readily explained by the loop capture mechanism.

Reviewer #1 (Significance (Required)):

*This study carefully quantifies chromosome shape parameters under mild cellular perturbations in *S. pombe*. The results indicate that chromosome shape is largely determined intrinsically, with minimal dependence on external physical factors, consistent with current understanding.*

We note that the current understanding, portrayed in the chromosome size literature, champions 'external physical factors' (e.g. Kimura et al 2013, Zhou et al. 2023). Our study was specifically designed to explore such factors, using a simple and genetically controllable model organism.

These findings will aid experts in chromosome biology in refining their models.

Reviewer expertise: condensin biology.

Reviewer #2 (Evidence, reproducibility and clarity (Required)):

Summary:

*-----
Provide a short summary of the findings and key conclusions (including methodology and model system(s) where appropriate).*

*This study investigates how the dimensions (length and width) of mitotic chromosomes are determined. The authors used *Schizosaccharomyces pombe* (fission yeast) as a model organism and combined super-resolution microscopy with semi-automated quantitative analysis and genetic manipulations. Previous studies have proposed that cell volume, nuclear volume, and the duration of chromosome condensation influence the size of mitotic chromosomes. However, in the present work, the authors found no correlation between chromosome dimensions and either cell or nuclear size, nor did they detect any effect of condensation duration. Instead, the analyses revealed a clear correlation between the amount of condensin complex and chromosome dimensions: reducing condensin levels led to an increase in chromosome width and length, whereas increasing condensin reduced them. These results suggest that the intrinsic amount of condensin, rather than extrinsic cellular or temporal factors, plays a key role in determining mitotic chromosome dimensions in fission yeast.*

We thank the reviewer for an accurate appraisal of our work.

Major comments:

*-----
- Are the key conclusions convincing?*

1. Use of arrested cells for chromosome dimension measurements:

All measurements were performed in synchronized populations-metaphase-arrested cells for chromosome width and G2-arrested cells for chromosome length. The authors should provide evidence that the chromosome dimensions measured under these arrest conditions are comparable to those in normally cycling cells, or discuss the rationale for assuming that the arrested conditions do not distort chromosome morphology.

As suggested by the reviewer, as part of our revision experiments, we turned to asynchronously growing cell populations and measured chromosome dimensions in the small, naturally occurring

fraction of mitotic cells. Reassuringly, chromosome arm lengths, measured in easily identifiable anaphase cells, were indistinguishable from those measured during synchronous anaphase following G2 arrest and release. Metaphase cells are harder to pinpoint in asynchronously growing populations. We used a tubulin-GFP marker to aid the identification of cells containing short mitotic spindles. Chromosome widths measured in such short-spindle cells were on average narrower than those observed in metaphase arrested cells. The difference could be caused by (i) pro-metaphase cells in our asynchronous population, which already display spindles but have not yet fully compacted their chromosomes. Alternatively, (ii) chromosomes might indeed reach a greater level of compaction during a mitotic arrest, when compared to unimpeded mitotic progression. These results are shown in a new Figure 5 and are discussed in the main text.

2. Uncertainty in measuring chromosome length:

It is unclear whether chromosome length was accurately traced in all cases.

In Figure S2B, the thicker chromosomal segment appears traceable, but the thinner or faint structures are difficult to follow precisely, raising concerns about endpoint determination. To better evaluate the validity of the tracing approach, the authors should include examples showing the tracing of all six chromatids in Fig. S2B.

In addition, they should discuss how reliably the chromosome endpoints-especially those near the spindle poles-can be determined and how this affects the accuracy of the measured lengths.

We have added a methods section that describes our chromosome arm length measurements. All discernible arms, with clearly identifiable beginnings and ends, and pointing in the direction of anaphase separation, were included in the analysis. Typically, two to three arms were measurable in this way per spindle pole. As the DNA stain does not distinguish between the six fission yeast chromosome arms, the measurements are an aggregate of the lengths of the six arms.

3. Uncertainty in width measurements:

In the current data, although statistical tests yield significant p-values, the plotted distributions appear to overlap extensively, suggesting that the differences are subtle at best.

Therefore, it remains unclear whether these measurements can robustly detect (or exclude) effects of cell size, nuclear size, or condensation time. The authors should more clearly justify the accuracy and sensitivity of their width measurements and explain why the chosen statistical tests are appropriate given the variability and potential measurement uncertainty.

The reviewer raises an important limitation of approaches to characterise chromosome dimensions. Overlapping width and length distributions were generally encountered in previous studies that measured chromosome dimensions in both yeasts and vertebrates (e.g. Kimura et al 2013, Kakui et al. 2022, Zhou et al. 2023). Despite these overlaps, statistical analysis of the size distributions allows us to distinguish between parameters that do or that do not result in chromosome size distribution changes.

4. Interpretation of the condensation time experiment (Figure 4B)

In the analysis of condensation time, even the shortest treatment corresponds to approximately 3 hours of mitotic arrest, which may already represent a saturated state of chromosome condensation. To more directly assess the possible time dependence, it would therefore be most informative to include a zero-time (no-drug) condition as a reference, as noted above in comment 1. Such data would help determine whether chromosome dimensions are indeed unaffected by condensation duration or whether the effect is simply saturated under all tested conditions.

This point relates to comment 1. Please see above for our response how we used chromosome measurements in asynchronously growing cells to address this concern.

- Should the authors qualify some of their claims as preliminary or speculative, or remove them altogether?

5. Unless the issues raised above are adequately addressed, the corresponding conclusions should be toned down.

We have tried our best to address the above issues, and we have also revised the text. Furthermore, we have added a 'Limitations of this study' section to the end of our discussion, that highlights the limitations of our study.

- *Would additional experiments be essential to support the claims of the paper? Request additional experiments only where necessary for the paper as it is, and do not ask authors to open new lines of experimentation.*

6. Need for validation of chromosome dimension measurements in non-arrested cells

As related to concerns 1 and 4 above, it would be important to measure or at least estimate the length and width of mitotic chromosomes in non-synchronized (non-arrested) cells. Such data are essential to demonstrate that the values obtained under arrest conditions are representative of normal physiology.

See above, points 1 and 4

7. Since the measurement of chromosome dimensions (length and width) is the central theme of the study, an independent imaging or quantification method-would substantially strengthen the credibility of the findings.

 - *Are the suggested experiments realistic in terms of time and resources? It would help if you could add an estimated cost and time investment for substantial experiments.*

7. I am not familiar with which experimental approaches are most effective in Schizosaccharomyces pombe. However, is it possible to perform chromosome-spread preparations to quantify chromosome dimensions? It would be helpful if the authors could consider whether established methods used in other organisms could be adapted for fission yeast. If no realistic alternatives exist beyond the current approach, the authors should at least discuss why conventional methods-such as chromosome spreading-are technically or biologically difficult to apply in this system.

As suggested by the reviewer, we have performed chromosome-spread preparations. The chromosome spreading protocol uses paraformaldehyde as an alternative fixative, and we expect that spread chromosomes are more readily measurable. Indeed, spread chromosome width measurements yielded comparable numbers to those obtained from our whole mount samples. However, we noticed slide-to-slide variation between the more than one slides that we prepared from each sample. The chromosome spreading protocol therefore appears to introduce a tendency to distort fission yeast chromosome width. To avoid such distortion, we decided to perform most measurements on whole mount samples, where chromosomes are protected from distortion, and where we did not observe slide-to-slide variation. We included an example chromosome spreading result in a new Figure S4B.

 - *Are the data and the methods presented in such a way that they can be reproduced?*

8. *yes*

 - *Are the experiments adequately replicated and statistical analysis adequate?*

9. As far as I can tell, the statistical analyses appear to have been performed appropriately. However, since I am not a statistician, I may not be in a position to fully assess their validity. When looking at the plots, the differences between conditions are not visually obvious, and the interpretation therefore relies heavily on the statistical testing. Given the large sample sizes, the reported p-values are not particularly small, raising doubts as to whether the observed differences are truly meaningful. The authors should clarify how they interpret these results and whether the statistical outcomes indeed support the stated conclusions.

We revisited all quantitative comparisons and the statistical tests that we employed. We accept that some measured differences, while significant, appear small. We note that a 2-fold chromosome volume difference manifests as only a 1.26 (cubic-root of 2)-fold length or width difference. We added a new 'limitations of this study' section to the discussion, in which we discuss this and other limitations of our study.

Minor comments:

 - *Specific experimental issues that are easily addressable.*

10. In the analysis of condensation time, it may be more informative to evaluate changes in chromosome length rather than width. Length measurements appear to show clearer differences and, importantly, do not require metaphase arrest. Assessing condensation kinetics based on chromosome length would therefore provide a more direct and less perturbative evaluation.

The reviewer makes an excellent suggestion, measuring anaphase chromosome lengths after increasing times of mitotic arrest. During our revisions, we have attempted to perform this experiment. Mitotic arrest is achieved by thiamine-induced repression of the Slp1 cell cycle regulator under control of the *nmt41* promoter. To resume mitotic progression, we must re-activate the *nmt41* promoter. However, while *nmt41* promoter repression by thiamine addition is fast, re-activation after thiamine removal is known to be slow (taking 14 - 20 hours). This slow response unfortunately precluded us from successfully and synchronously releasing mitotically arrested cells for monitoring chromosomes during anaphase.

 - *Are prior studies referenced appropriately?*

11. *yes*

 - *Are the text and figures clear and accurate?*

*12. Overall, the text and figures are clear and well presented. However, as noted above, it seems technically challenging to measure mitotic chromosome dimensions accurately in *Schizosaccharomyces pombe*. Given these inherent difficulties, the rationale for choosing fission yeast as the model organism for this type of analysis is not entirely clear. The authors should explicitly explain why *S. pombe* is advantageous or appropriate for studying chromosome dimensions in this context.*

Yeast models offer exquisite genetic control over cellular variables, like cell and nuclear size and cell cycle stage. Amongst the yeast models, in turn, fission yeast has three, relatively large chromosomes. These considerations motivated our model organism choice. At the same time, we concede that yeast mitotic chromosomes are harder to image than those of higher eukaryotes, a drawback that we discuss in our newly added 'limitations of this study' section.

 - *Do you have suggestions that would help the authors improve the presentation of their data and conclusions?*

13. I suggest the authors to include schematic timelines in the figures to indicate when drug addition/removal and image acquisition occurred in each experiment. This would help readers easily understand the time course and treatment conditions associated with each dataset.

Thank you for this suggestion, which we appreciate. In response, we created figures with added experimental schematics. However, instead of the expected clarity, the schematics added complexity. After consultation with colleagues, we decided to forego on the schematics. Instead, we revisited all figure legends to ensure that they clearly describe the experimental workflow. We hope that this course of action is acceptable.

****Referees cross-commenting****

I have read the other two referee reports and found them reasonable and well balanced. Some of their comments overlap with my own, and even those that do not appear well justified and worth addressing by the authors.

Although it may be difficult to perform all the requested experiments, I strongly encourage the authors to discuss the methodological limitations and how these might affect the interpretation of their conclusions. Providing a transparent and balanced discussion of these aspects would substantially strengthen the manuscript.

Regarding the novelty of the study, I think the concern raised by another reviewer is valid. At the same time, if the robustness of the data can be further strengthened and the authors provide a

thorough discussion of the relevant prior work while positioning their findings fairly within that context, the paper would hold value for the field.

We added additional data in support of our conclusions, and revised the manuscript to provide a transparent and balanced discussion, including a new 'limitations of this study' paragraph.

Reviewer #2 (Significance (Required)):

- Describe the nature and significance of the advance (e.g. conceptual, technical, clinical) for the field.

The finding that condensin influences the dimensions (width and length) of mitotic chromosomes is conceptually not entirely surprising, but the authors' demonstration of this relationship through direct in vivo manipulation is valuable and represents a meaningful technical contribution to the field.

However, while the study also concludes that chromosome dimensions are independent of cell size, nuclear size, and condensation time, these negative results should be interpreted with caution. Given the uncertainties in measurement accuracy discussed above, the data may not be sufficient to definitively exclude such influences. The authors are encouraged to moderate these claims accordingly.

- Place the work in the context of the existing literature (provide references, where appropriate).

As far as I can tell, the Introduction accurately and appropriately summarizes the relevant literature. The authors provide a clear overview of previous findings on chromosome dimensions (length and width) across different species and the molecular factors implicated in these processes.

Importantly, while various studies have described correlations between chromosome morphology and specific proteins, there appear to be very few examples in which the effects of condensin on chromosome dimensions have been directly manipulated and quantified in vivo. Therefore, this study makes a meaningful contribution by providing direct experimental evidence on this point.

- State what audience might be interested in and influenced by the reported findings.

The work will be of interest to researchers studying chromosome architecture and organization. It may also attract attention from those interested in cellular- and nuclear-scale size control mechanisms, even beyond the field of chromosome biology.

- Define your field of expertise with a few keywords to help the authors contextualize your point of view. Indicate if there are any parts of the paper that you do not have sufficient expertise to evaluate.

*I am a cell biologist with a particular interest and research focus in size control mechanisms. I do not have direct experimental experience with *Schizosaccharomyces pombe*. While I have general knowledge of statistical analysis, I am not an expert in the specific statistical approaches required to rigorously evaluate subtle quantitative differences such as those analyzed in this study.*

Reviewer #3 (Evidence, reproducibility and clarity (Required)):

Summary

Wu and colleagues quantified chromosome area, arm width, and arm length in fission yeast by measuring DAPI-stained chromatin in cells released into mitosis after varying durations of G2-phase arrest. During prolonged G2 arrest, chromatin area expanded continuously, in proportion to increases in nuclear and cell size. Upon entry into mitosis, chromosomes re-compacted to a similar extent, restoring arm width, but not arm length, regardless of the length of G2 arrest.

In mutant strains with smaller cell size, mitotic chromosomes likewise showed no major differences in occupied area or arm width but exhibited somewhat increased arm lengths. Western blot analysis indicated that the concentration of the condensin subunit Cut14 (SMC2) was reduced both, in larger cells following prolonged G2 arrest and in smaller mutant cells. Gradual depletion of Cut14 led to a corresponding increase in metaphase chromosome arm width and, when Cut14 levels were reduced to roughly half, an increase in anaphase chromosome length. Conversely, co-overexpression of the five condensin subunits caused a slight decrease in metaphase arm width and anaphase chromosome length.

The authors conclude that mitotic chromosome compaction depends on the cellular level of condensin, but is independent of cell or nuclear size.

We thank the reviewer for an accurate appraisal of our work.

Major comments

1. The authors conclude that reduced condensin levels in larger or smaller cells limit chromosome arm shortening (p. 8), based on Western blot analyses of a single condensin subunit (Figs. 1D and 2D). However, this interpretation would only be valid if Cut14 were the rate-limiting component of the five-subunit condensin complex; an assumption that seems unlikely.

Condensin is a stoichiometric five-subunit protein complex, and each of its five subunits is essential for its function in fission yeast. Therefore, reducing the levels of any one of the five components, especially one of the core ATPase subunits, results in a corresponding reduction of functional condensin complexes.

Furthermore, even if Cut14 were rate-limiting, the manuscript does not demonstrate that a 10-30 % reduction in total cellular Cut14 levels leads to a corresponding decrease in condensin abundance on chromosomes, as claimed in the abstract ("Instead, levels of the chromosomal condensin complex..."). In the Discussion, the authors acknowledge that quantitative assessment of chromosome-bound condensin is "experimentally challenging." Nonetheless, established methods such as spike-in ChIP-seq offer a feasible and quantitative approach to measure chromatin-associated SMC protein levels genome-wide (Hu et al., Nucleic Acids Res., 2015).

We agree with the reviewer that establishing a robust condensin ChIP approach to directly measure chromosome-bound condensin levels would be informative. However, this approach goes beyond the practicable scope of our revisions. Unlike cohesin, studied by Hu et al. 2015, condensin is a much trickier target for (quantitative) ChIP analyses.

2. The authors use "chromatin area" as a measure of chromosome compaction. However, it is unclear from the Methods section whether this area was derived from a maximum-intensity projection of the DAPI signal or from a single z-slice. In either case, quantifying chromatin volume would seem more appropriate for assessing chromosome compaction. Additionally, the use of a 500 nm z-spacing raises concerns, given that the fission yeast nucleus has a diameter of only about 1 μm .

Indeed, we used maximum intensity projections as the input for the ilastik tool to segment the chromatin-occupied area. This information has been added to the methods section. Most of the currently available segmentation tools operate on 2D images. While 3D segmentation is possible (e.g. Stamatov et al. 2025), it would be much more demanding and likely less accurate.

3. Another concern relates to the measurement of chromosome length in anaphase cells. It is unclear how the authors accurately identified the centromere position of individual chromosomes within clustered chromatids. The Methods section lacks a description of how chromosome arm lengths were determined.

Apologies for this oversight. We have added a methods section that describes our chromosome arm length measurements. All discernible arms, with clearly identifiable beginnings and ends, and pointing in the direction of anaphase separation, were included in the analysis. Typically, two to three arms were measurable in this way per spindle pole. As the DNA stain does not distinguish between the six fission yeast chromosome arms, the measurements are an aggregate of the lengths of the six arms.

Previous studies have employed fluorescently labeled reporter arrays to precisely measure chromosome lengths in yeast species, including fission yeast (e.g., Sakuno et al., Nature 2009;

Petrova et al., Mol. Cell Biol. 2013). Given that the main conclusions of this work rely on very small differences in chromosome length (see next comment), it is essential that the authors validate their findings using such established, high-resolution approaches available for their model system. Moreover, the authors should consider that apparent differences in anaphase chromosome arm length may reflect stretching by microtubule pulling forces. The use of reporter arrays would enable measurements in metaphase-arrested cells, mitigating this concern.

Following the reviewer's suggestion, we contemplated using one of the previously developed reporter array systems to measure chromosome formation, but decided against their use for the following reasons. The reporters mark two loci at shorter distance than chromosome arm length, so any small chromosome arm length changes would be manifest in even smaller array distances changes. More importantly, while array distances can be measured accurately, the chromosome path between the two loci, and with it the chromosome contour length, remains unknown.

4. Although many of the reported differences reach statistical significance, the corresponding effect sizes are often small. For instance, the measured chromosome arm lengths differ by only 0.1-0.15 μm , less than 10 % of the total arm length (Figs. 1C, 2C, and 3C). The reduction observed upon condensin overexpression is even smaller, at approximately 0.05 μm (Fig. 4C). Notably, the authors describe this minor difference as "significantly shorter" ($p = 0.03$), yet refer to the reduction in chromosome width in the ppa2 mutant as "marginally reduced" (Fig. 2B), despite a comparable level of statistical significance ($p = 0.02$).

We thank the reviewer for the prompt to ensure the use of accurate and consistent descriptions when reporting the results of our chromosome measurements. We have modified the manuscript text accordingly. Additionally, we have added a 'limitation of this study' section at the end of our discussion, which highlights the difficulties with accurately measuring yeast chromosomes.

5. Assessing the statistical significance of Western blot signals (Figs. 1D, 2D, and 3C) based on a sample size of $n = 3$ is questionable, particularly if signal linearity cannot be ensured when using enzyme-linked secondary antibodies (e.g., HRP), as signal saturation may occur. The Methods section does not specify what steps were taken by the authors to verify linearity or to ensure accurate signal quantification.

Western blotting is a widely used, quantitative method to assess protein levels. We ensured that all quantified signals were in the linear detection range, and we performed all quantifications in three independent biological repeat experiments. Details of our approach have been included in an additional methods section.

As an alternative approach, the authors could consider fluorescently tagging condensin subunits and comparing total fluorescence signals between strains to obtain a quantitative measure of protein levels.

Fluorescence signal quantification is a great alternative method to measure protein levels, which however comes with its own confounding issues relating to signal quenching, fluorophore bleaching and background correction.

Minor comments

1. It is unclear why the authors arrested cells and depleted Cdc14 by auxin addition for 5 hours to measure metaphase chromosome arm widths (Fig. 3B), but only for 3 hours (plus the time until anaphase) to measure anaphase chromosome lengths (Fig. 3C). This discrepancy likely accounts for the observed difference in Cdc14 depletion levels-approximately 10 % remaining at 500 μM IAA in the former experiment versus about 50 % in the latter. To enable a more meaningful comparison between the two conditions, the authors could consider shortening the Slp1 arrest to 3.5 hours, which should allow complete metaphase arrest while achieving condensin depletion levels comparable to those in the anaphase release experiment.

5 hours depletion were used in Figure 3B to reach a wide dynamic depletion range, and the reviewer is correct that similar depletion levels were inaccessible within only 3 hours. The 3-hour limit in the experiment shown in Figure 3C was imposed by the practicalities of synchronous anaphase release of these cells.

2. The authors contend that the observed shortening of chromosome length upon condensin overexpression supports a loop capture model rather than a loop extrusion model. However, under a loop extrusion mechanism, an excess of condensin would be expected to produce smaller loops

and, consequently, longer chromosome arms only if colliding condensin complexes were unable to bypass one another, which does not appear to be the case (Kim et al., Nature 2020).

Whether condensins bypass each other, were they to extrude chromatin loops *in vivo*, is currently a topic of debate in the loop extrusion field. Irrespective of bypass, the loop extrusion model suggests that fewer condensins generate fewer and longer loops, resulting in wider and shorter chromosomes. This prediction is contrary to our observation that chromosomes become wider and longer.

3. The authors propose that alterations in transcriptional regulation may underlie the reduced cell and nuclear size observed upon condensin depletion, citing their 2021 publication. However, they do not acknowledge previous evidence demonstrating that condensin does not play a direct role in transcriptome maintenance in fission yeast (Hocquet et al., eLife, 2018). This apparent discrepancy should be explicitly discussed.

We thank the reviewer for pointing us to the Hocquet et al. paper. These authors document gene expression changes that are due to cell division errors in the absence of condensin. The authors do not, however, study condensin-dependent gene expression changes in the absence of cell divisions (apart from 3 selected ‘cell division-affected’ genes). We include a reference to the Hocquet et al. study in our revised manuscript.

Reviewer #3 (Significance (Required)):

The question of which factors shape mitotic chromosomes has long been a central issue in cell biology. Foundational insights into this problem, particularly the pivotal role of condensin in organizing mitotic chromosomes, have come from experiments using Xenopus egg extracts. However, this in vitro system does not allow for assessing the influence of cell or nuclear size on mitotic chromosome architecture. The demonstration that mitotic chromosomes do not scale with cell or nuclear size in fission yeast therefore represents a notable contribution.

Nonetheless, the observation that chromosome length is determined by condensin is largely consistent with previous findings and thus not entirely unexpected. Previous studies have shown that anaphase chromosome length increases in fission yeast condensin mutants (Petrova et al., Mol Cell Biol 2013), including strains in which expression of the condensin subunit Cnd1 is reduced by half (Schiklenk et al., J Cell Biol 2018). Surprisingly, none of these studies are cited in the manuscript. Instead, the majority of references point primarily to the authors' own previous work.

Apologies for overlooking these additional, relevant, papers from the Haering lab. These papers confirm, and quantitatively underscore, what was previously established by work in the Yanagida lab (that we referred to). We have included references to these additional studies in our revised manuscript. We also note that our study goes beyond previous approaches that have characterised the consequences of condensin reduction. Only by increasing the condensin dosage, and by observing resultant chromosome hyper-compaction, can we conclude that condensin is indeed a rate limiting component for chromosome formation.

In conclusion, while the study offers interesting observations, its overall novelty appears limited, and several experimental concerns remain unresolved. Substantial additional experimentation would be needed to fully support the conclusions, and these revisions may extend beyond what is feasible for a standard revision.

Second decision letter

MS ID#: jcs.264569R1

MS Title: Investigations into fission yeast chromosome size determinants

Authors: Pei-Shang Wu; Todd Fallesen; Frank Uhlmann

Article Type: Review Commons Transfer

Dear Frank,

Happy New Year and thank you for your resubmission. I am happy to tell you that your manuscript has been accepted for publication in Journal of Cell Science, pending standard publication integrity checks.

Thank you for sending your manuscript to Journal of Cell Science through Review Commons.